

# Flatfoot in Africa, the cirripede *Chthamalus* in the west Indian Ocean

Noa Simon-Blecher[1], Avi Jacob[1], Oren Levy[1], Lior Appelbaum[1,2], Shiran Elbaz-Ifrah[1] and Yair Achituv[1]

[1] The Mina & Everard Goodman Faculty of Life Sciences, Bar-Ilan University, Ramat-Gan, Israel
[2] The Multidisciplinary Brain Research Center, Bar-Ilan University, Ramat Gan, Israel

## ABSTRACT

Barnacles of the genus *Chthamalus* are commonly encountered rocky intertidal shores. The phylogeography of the different species in the Western Indian Ocean is unclear. Using morphological characteristics as well as the molecular markers mitochondrial cytochrome oxygenase subunit I (COI) and the nuclear sodium-potassium ATPase (NaKA), we identified four clades representing four species in the Western Indian Ocean and its adjacent seas. Among these species, a newly identified species, *Chthamalus barilani*, which was found in Madagascar, Zanzibar and Tanzania. *Chthamalus* from the coasts of Tanzania and Zanzibar is identified morphologically as *C. malayensis*, and clusters with *C. malayensis* from the Western Pacific and the Indo Malayan regions. *C. malayensis* is regarded as a group of four genetically differentiated clades representing four cryptic species. The newly identified African clade is genetically different from these clades and the pairwise distances between them justify the conclusion that it is an additional cryptic species of *C. malayensis*. This type of genetic analyses offers an advantage over morphological characterization and allowed us to reveal that another species, *C. barnesi*, which is known from the Red Sea, is also distributed in the Arabian Sea and the Persian Gulf. We could also confirm the presence of the South African species *C. dentatus* in the Mozambique channel. This represents the Northeastern limit of *C. dentatus*, which is usually distributed along the coast of southern Africa up to the Islands of Cape Verde in West Africa. Altogether, based on a combination of morphology and genetics, we distinct between four clusters of *Chthamalus*, and designate their distribution in the West Indian Ocean. These distinctions do not agree with the traditional four groups reported previously based merely on morphological data. Furthermore, these findings underline the importance of a combining morphological and genetics tools for constructing barnacle taxonomy.

## INTRODUCTION

Barnacles of the genus *Chthamalus* are some of the most commonly encountered intertidal barnacles with a worldwide distribution of rocky intertidal shores. In his monumental monograph on extant acorn barnacles, *Darwin (1854)* recognized eight species of barnacles

Corresponding author
Yair Achituv, achity@gmail.com

of the discrete genus *Chthamalus Ranzani, 1817*. Some of them, like *Chthamalus stellatus Poli, 1791*, encompass a number of races or varieties that were later recognized as valid species, including new species of *Chthamalus*. This genus has the widest distribution in the family of Chthamalidae; currently, 27 nominal species of *Chthamalus* are recognized (*Chan et al., 2021*) The morphological and ecological similarities within the genus may give rise to some taxonomic confusion. However, the use of molecular techniques has resolved some of the uncertainty and enabled the identification of a number of cryptic species within long-established nominal species. In some instances, such distinctions were subsequently confirmed by taxonomists using morphological parameters. In the present study we used both molecular techniques and morphology to describe the West Indian Ocean (WIO) populations of *Chthamalus*.

Our knowledge of the cirripedes, and particularly of the Chthmaloidea of the West Indian Ocean, is limited. *Broch (1927)* based his report of the presence of *Chthamalus challengeri Hoek, 1883* in the Red Sea, on three specimens collected by the Cambridge expedition to the Suez Canal. However, there have been no further observations of this species in the Red Sea or on the East coast of Africa. *Achituv & Safriel (1980)* described a new species, *Chthamalus barnesi Achituv & Safriel, 1980*, from the Red Sea, which is also found in the Gulf of Suez, the Gulf of Aqaba, and the Dahlak Archipelagos, but its global distribution is not known. *Southward & Newman (2003)*, who reviewed the taxonomy and distribution of the Indo Malayan and West Pacific barnacles of the Genus *Chthamalus* from the East Africa in the west to Queensland, Australia in the east and the Ryukyu Islands in the north, recorded three species, namely *Chthamalus malayensis Pilsbry, 1916*, *Chthamalus challengeri*, and *Chthamalus moro Pilsbry, 1916*. They noted that the taxonomic status of *Chthamalus* spp. from the West Indian Ocean (WIO), is unclear but grouped it under *C.* cf. *challengeri*. While their review does not include the Red Sea, the Persian Gulf, or the shores of the Arabian Sea, *Shahdadi & Sari (2011)* reported the existence of *C. barnesi* in the Persian Gulf and the Gulf of Oman.

*Ren (1989)* described the barnacles collected in Madagascar by Dr. Alain Crosnier during 1956–1975 when he was on the staff of ORSTOM (Office de la recherché Scientifique et Technique Outre-Mer) in Nosy Be. The collection contains 28 species of barnacles collected from the intertidal zone to a depth of about 2,000 m, but does not include any *Chthamalus*, although *Newman & Ross (1976)* listed Madagascar within the range of distribution of *Chthamalus dentatus Krauss, 1848*. *Southward & Newman (2003)* refer to *Chthamalus* from Madagascar as "probably *C. malayensis*".

Traditionally, the identification of barnacles is based on morphological features, although several taxa (from the entire class Thecostraca to subfamilies) are essentially only recognized by molecular analyses (*Chan et al., 2021*). In the field, where only the morphology of the shell and opercular valves can be used for identification, the high variability in some species of *Chthamalus*, may give rise to confusion and lead to the misclassification of morphologically different species (*Southward, 1976*). As a corollary, morphological similarity may lead to the mistaken consolidation of different species (*Wares, 2020*). This becomes even more challenging in locations with sympatric species. In addition, predation, exposure to erosion and differential growth (mainly of opercular

valves) may lead to morphological variation within species (*Lively, 1986*; *Foster, 1974*). Although a consideration of the morphology of the segmented appendages, cirri, and trophi is often used to distinguish variants. The developing field of molecular biology has become a major component in taxonomic research, including barnacles (*Pitombo & Burton, 2007*). DNA sequences and specifically those of the mitochondrial gene encoding Cytochrome c oxidase subunit 1 (COI) may be useful to distinguish taxa. For example, *Chan & Cheang (2015)*; *Chan et al., 2016*) used enzyme electrophoresis and the COI gene to identify seven chthamalid species from the East Pacific. *Pérez-Losada, Høeg & Crandall (2004*; *Pérez-Losada et al., 2012*, *2014*) included species of *Chthamalus* in these studies (5 in 2004; 10 in 2014) on the radiation of the thoracican barnacles.

In the present study, we describe the biogeographic pattern of the genus *Chthamalus* in the West Indian Ocean and the adjacent Arabian sea, the Oman Gulf, the Persian Gulf, and the Red Sea. The examination of barnacles collected in these seas revealed four *Chthamalus* species including a new species.

## MATERIALS AND METHODS

Samples of *Chthamalus* collected by us from the intertidal rocks or donated by colleagues were fixed and stored in 96% ethanol. The material used for the present study is stored in the Israeli National Natural History Collections at the Hebrew University of Jerusalem (for details, see Supplemental Material 1). Collection sites are presented in Fig. 1.

Three-dimensional figures of selected specimens were visualized by Olympus SZX10 dissecting microscope; figures were combined using CellSense software (Fig. 2). Hard parts were prepared for SEM examination using the method described by *Achituv & Hoeksema (2003)* and recently used by *Tikochinski et al. (2020)*. Shell and opercular plates were separated from individual barnacles, soaked in 5% sodium hypochlorite (household bleach) for 2 h, and then examined under the stereo microscope. Adherent chitin and foreign debris were then removed using entomological needles and a fine paintbrush. Dried samples were mounted on brass stubs, coated with gold, and examined with a JEOL scanning electron microscope at 25 kV. Images were stored using Autobeam software.

Images of soft parts, cirri, and mouthparts of selected specimens were examined and photographed with a Leica upright field LMD7 microscope. The figures were acquired with LasX software. Setae of the first and second cirri were dehydrated using the critical point drying (CPD) method, mounted on brass stubs, and examined and photographed by SEM at 25 kV with Autobeam software.

The electronic version of this article is Portable Document Format (PDF) will represent a published work according to the International Commission on Zoological Nomenclature (ICZN), and hence the new names contained in the electronic version are effectively published under that Code from the electronic edition alone. The published work and the nomenclatural acts it contains have been registered in Zoobank, the online registration system for the ICZN. The ZooBank LSIDs (Life Science Identifiers) can be resolved and the associated information viewed through any standard web browser by appending the LSID to the prefix http://zoobank.org/. The LSID for this publication is urn:lsid:zoobank.org:pub:0362DDF4-D56E-4431-9639-3DBF772E221C. Publication LSID: urn:lsid:zoobank.
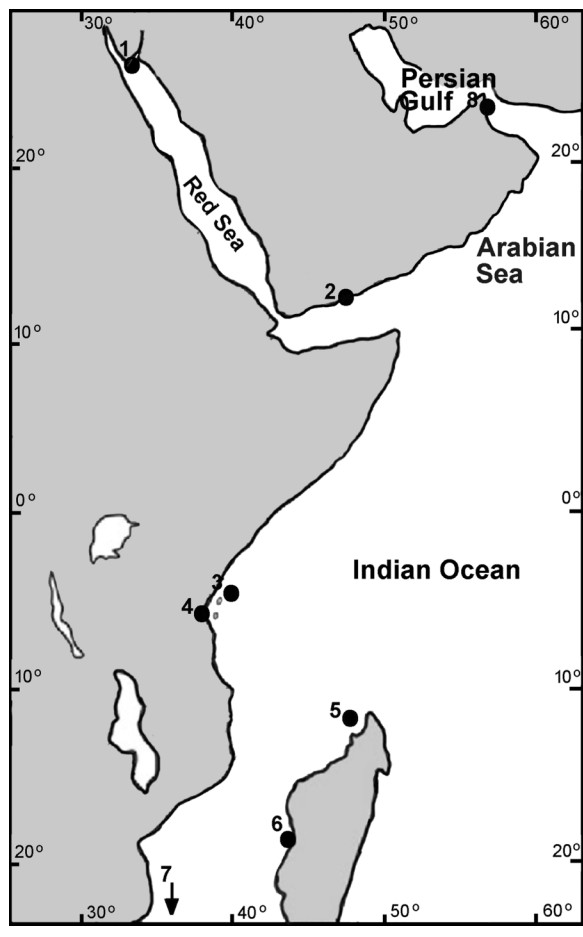

**Figure 1 Collection sites of Chthamalus in the West Indian Ocean.** Collection location of Chthamalus in the WIO 1. Nabeq, Red Sea. 2. Yemen, Arabian Sea. 3. Zanzibar, Tanzania. 4. Dar es Salaam, Tanzania. 5. Nosy Be, Madagascar. 6. Morondava, Madagascar. 7. Durban, South Africa (not on the map). 8. Strait of Hormuz, Persian Gulf.

org:pub:0362DDF4-D56E-4431-9639-3DBF772E221C. The online version of this work is archived and available from the following digital repositories: PeerJ, PubMED Central and CLOCKSS.

DNA was extracted from ethanol fixed muscles and cirri of barnacles using a genomic DNA isolation kit (MACHEREY-NAGEL GmbH & Co., Germany), following the manufacturer's protocol. The concentration of DNA was determined by NanoDrop ND1000 (Thermo Fisher Scientific Inc., Waltham, MA, USA) at 260 nm.
The mitochondrial Cytochrome Oxygenase Subunit I (COI) and a partial of the coding sequence of the nuclear gene sodium-potassium ATPase (NaKA) were used as molecular markers. The COI was amplified by the universal barcode primers LCO1490 and HCO2198 (*Folmer et al., 1994*). For the amplification of NaKA we used the primer of *Wares et al. (2009)* F: 5′-GTGGTTCGACAACCAGATCA; R: 5′-GGGATCTCGCA GACCTTCTT. The amplifications were conducted, as previously described in *Tikochinski et al. (2021)*, The reaction mix containing 0.5 μl of template DNA, 12.5 μl PCR mix (2X PCR HS Taq Mix Red; PCRBiosystem, London, UK) 10 nM of each primer, and double

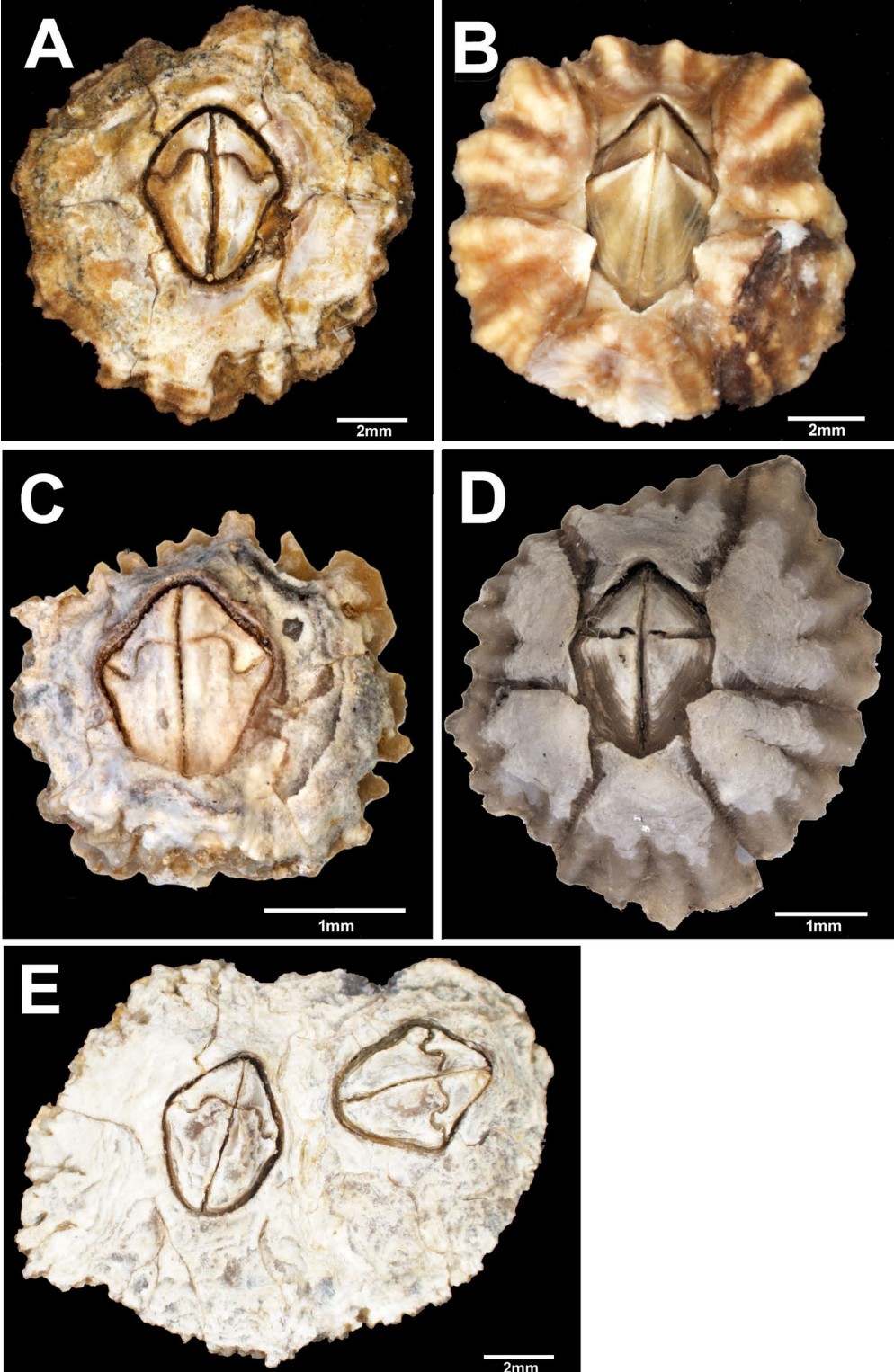

**Figure 2 External view of *Chthamalus* from Eastern Indian Ocean.** External view of *Chthamalus* from Eastern Indian Ocean. (A) *C. barilani* from Nosy be, Madagascar, eroded specimen. (B) *C. barilani* n.sp. from Nosy Be, Madagascar non-eroded specimen. (C) *C. barnesi* from Ras Misela, Gulf of Suez, Egypt. (D) *C. dentatus* from Morondava, Madagascar. (E) *C. malayensis* from Dar E Salaam, Tanzania.

distilled H$_2$O to a total volume of 25 µl. The PCR profile for COI was 1 min at 95 °C for initial denaturation, then 40 cycles of 15 s at 95 °C, 15 s at 50 °C, 20 s at 72 °C with a final extension for 3 min at 72 °C 1 min at 4 °C and pause at 15 °C. For NaKA the profile was: 1 min at 94 °C for initial denaturation, then 35 cycles of 15 s at 94 °C, 15 s at 59 °C, 20 s at 72 °C with a final extension for 3 min at 72 °C 1 min at 4 °C and pause at 15 °C.

Amplification was carried out in a personal combi-thermocycler (Biometra, Germany). PCR products were purified and sequenced by MCLAB laboratories (San Francisco, California). Both strands were sequenced using an ABI PRISM 3100 Genetic Analyzer (Applied Biosystems, Foster City, CA, USA).

Sequences were translated aligned using Clustal X2 (*Larkin et al., 2007*). Phylogenetic analyses were performed based on maximum likelihood (ML) analysis, and 1,000 bootstrap replicates were conducted using MEGA7 (*Kumar, Stecher & Tamura, 2016*). Analyses were run under the best-fit models selected by MEGA7.

Posterior probabilities were estimated by Bayesian Inference using BEAST v1.10.4 (*Drummond et al., 2012*). We separately analyzed the COI and NaKA loci datasets. Posterior probabilities were generated in the BEAST (*Drummond & Rambaut, 2007*). The analyses were run for 10 million generations and sampled every 1,000 generations.

Sequences used in the present study have been deposited in GenBank; accession numbers are given in Supplemental Material 2.

## RESULTS

### Phylogeny

In addition to the sequences generated by us, we included the sequences of other species of *Chthamalus* found in the Indo West Pacific (IWP), as well as randomly selected sequences representing the four clades of *C. malayensis* identified by *Tsang et al. (2008*, *2012a)*. Two sequences of other Mediterranean species of *Chthamalus* were also added, and two species of *Octomeris, Octomeris brunnea Darwin, 1854* and *Octomeris angulosa Sowerby, 1825*, were used as an outgroup. The segments of COI used for analysis are 531 bp long, while those of NaKA are 183 bp

Figures 3 and 4 present the maximum likelihood phylogenetic trees based on COI and NaKA respectively. The model computed and selected by MEGA to construct the COI tree is General Time Reversible (GTR) + I (BIC = 10140; AICc = 8988); the analysis involved 156 nucleotide sequences. The tree with the highest log likelihood (−5076.9) is shown. For the NaKA phylogenetic tree, the Tamura 3-parameters model (T92) + G was selected (BIC = 3931; AIC = 1,902); the analysis involved 131 nucleotide sequences. The tree with the highest log likelihood (−699.1291) is shown. The reliability of the nodes in the phylogenetic trees was estimated using the bootstrap method in the ML analyses and the Bayesian posterior probabilities. These two methods of estimation were in good agreement for the COI tree but there was some discrepancy for the NaKA tree where the bootstrap support values are rather low while the posterior probabilities are high ranging between 0.91 and 1. *Douady et al. (2003)* who used experimental simulation of empirical data sets to compare the Bayesian posterior probabilities and ML bootstrap

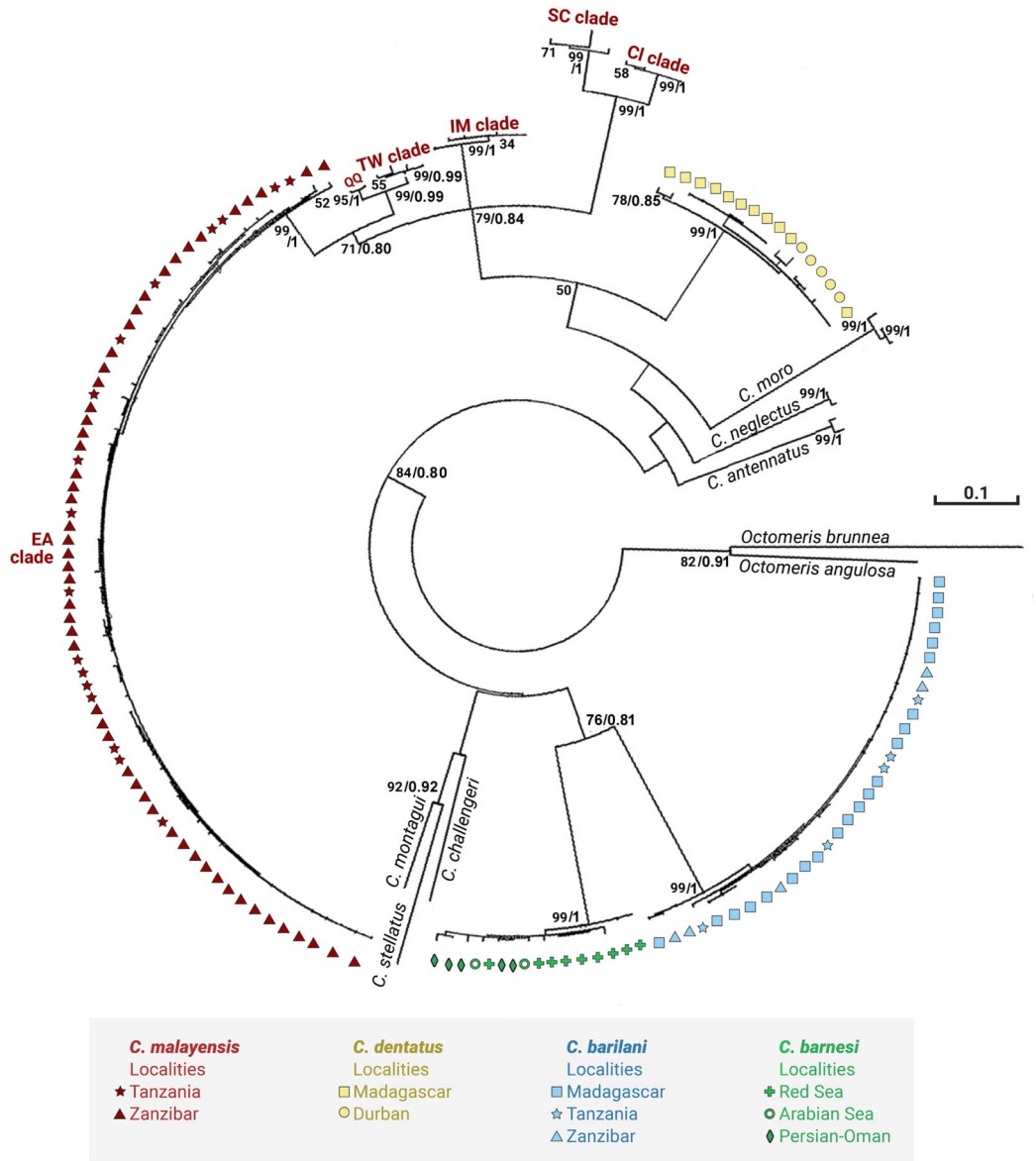

**Figure 3 Maximum likelihood phylogeny of *Chthamalus* from the West Indian Ocean based on COI gene sequences.** Maximum likelihood phylogeny based on COI gene sequences of *Chthamalus* from the West Indian Ocean generated in the present study and of selected species from other oceanic regions, retrieved from GenBank. Bootstrap values >50% and Bayesian posterior probabilities >0.75% are indicated.

supports concluded that posterior probabilities and bootstrapped maximum likelihood percentages cannot be directly compared.

The phylogenetic analyses based on the two markers revealed a similar phylogenetic pattern. Within the West Indian Ocean, we identified four main clades, each representing a different species. The COI sequences obtained from two clades show 93–99% identity to two known species of *Chthamalus*, namely *C. malayensis*, and *C. dentatus*. However, the sequences obtained from the other two clades are significantly different from all recorded chthamalid sequences in GenBank. The closest relation to other species of

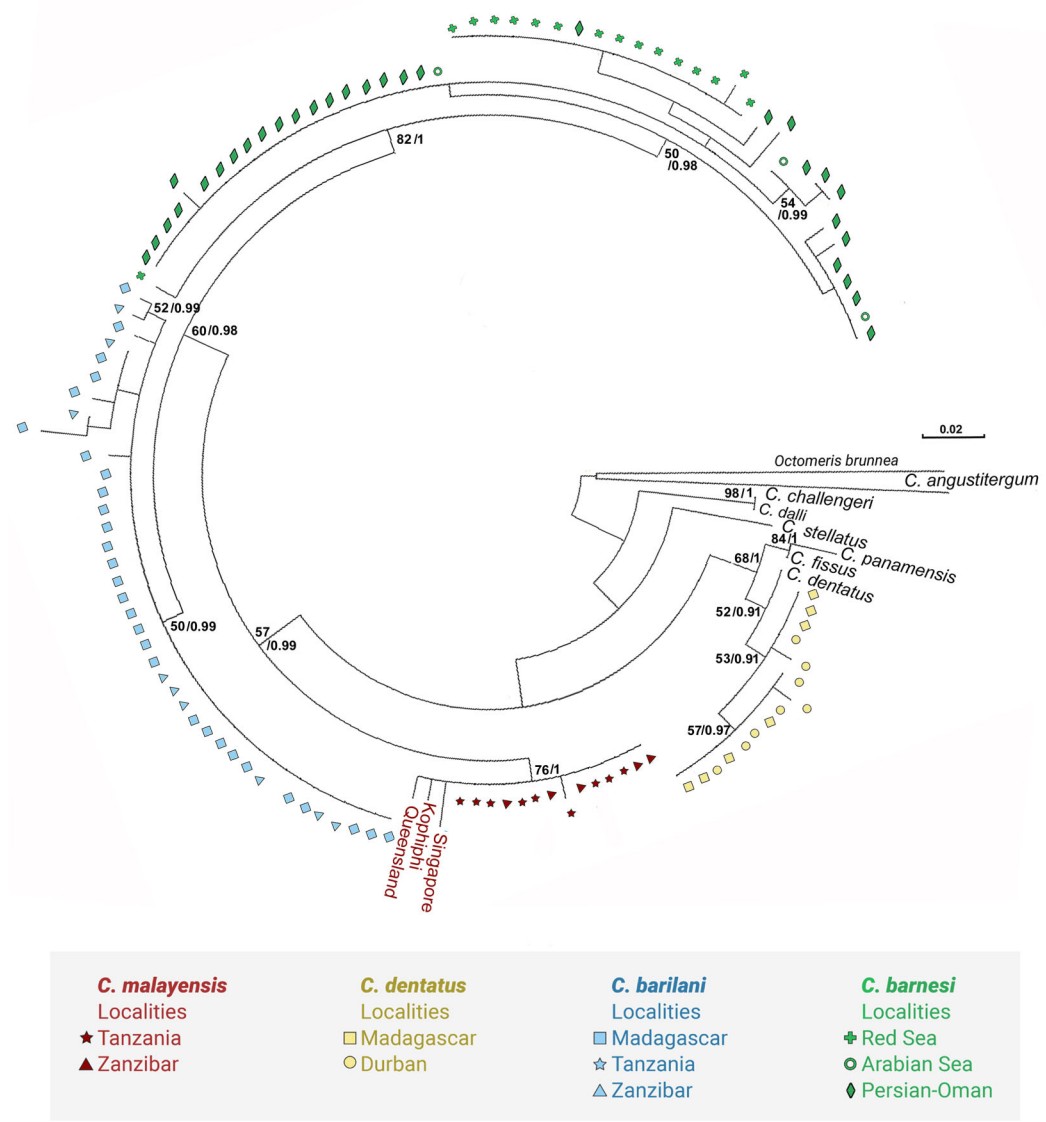

**Figure 4 Maximum likelihood phylogeny of *Chthamalus* from the West Indian based on NaKA gene sequences.** Maximum likelihood phylogeny based on NaKA gene sequences of *Chthamalus* from the West Indian Ocean generated in the present study and of selected species from other oceanic regions, retrieved from GenBank. Bootstrap values >50% and Bayesian posterior probabilities >0.75% are indicated.

*Chthamalus* is 88%, which is the difference found between different species of *Chthamalus*. One of the clades includes specimens that match the morphological description of *C. barnesi*. COI sequences that cluster in this clade were taken from specimens obtained from the sample used to describe *C. barnesi* (*Achituv & Safriel, 1980*). *C. barnesi* has been recorded from the Persian Gulf and the Gulf of Oman (*Shahdadi & Sari, 2011*). Samples from this area and other localities in the West Indian Ocean clustered with the samples from the Red Sea, enabling us to map the distribution of *C. barnesi*. The other independent clade contains sequences obtained from samples collected in Madagascar, Zanzibar, and Tanzania, and represents a new species we call *C. barilani*.

The pairwise distances between sequences of nominal species of *Chthamalus* (Table 1) range from 0,082 between *Chthamalus mexicanus Pitombo & Burton, 2007* and *Chthamalus alani* Chan 2016 to a maximum of 0.222 between *Chthamalus bisinuatus* (*Pilsbry, 1916*) and *Chthamalus montagui Southward, 1976*. The lowest pairwise distance value found between a randomly selected sequence from Madagascar and other species of *Chthamalus* is 0.151, which is within the range of distances among different species of *Chthamalus*. The highest value found, 0.228, is between *C. barilani* and *C. proteus Dando & Southward, 1980*. This justifies the allocation of *C. barilani* as a new species.

## Taxonomy

According to *Newman & Ross (1976)* and *Newman (1996)*, chthamaloids with a six plated shell, a quadridentate mandible, and complex setae on the second cirrus, belong to the genus *Chthamalus*. Identification of species or genera in the field usually depends solely on the morphology of the shell and the opercular valves. This can work well in some instances, for example, the two sympatric barnacles, *C. stellatus* and *Euraphia depressa* (*Poli, 1791*) can be easily separated using their external appearance. However, the high variability in the shell and opercular valves of some species of *Chthamalus* (*Foster, 1974*) and the similarities in morphology within and between other species, compounded by possible changes due to age and habitat greatly complicate the taxonomy of species of *Chthamalus*. *Achituv & Safriel (1980)* described prominent morphological differences in the shells and opercular valves of *C. barnesi* as a result of erosion. Non-eroded specimens have a low conical strongly ribbed shell, whereas in eroded specimens, the ribs are worn, the shell is tall, and the orifice is larger relative to shell diameter due to erosion of its circumference.

As a result, in many cases, and particularly in sympatric species, it may be impossible to distinguish species in the field. The specimens from Madagascar differ morphologically from those previously described in the West Indian Ocean. Nevertheless, as described for four species of chthamalids collected in Fiji, namely *Octomeris brunnea*, *Euraphia intertexta* (*Darwin, 1854*), *Euraphia caudate* (*Pilsbry, 1916*), and *Chthamalus malayensis*, barnacles genetically clustered in the same clade, and even those collected in the same locality, may display differences in the shells and opercular valves as a result of erosion (*Foster, 1974*).

### *Chthamalus barilani* Achituv Sp. Nov.
Family CHTHAMALIDAE *Darwin, 1854*
Genus *Chthamalus Ranzani, 1817*
*Chthamalus barilani* sp. nov. Achituv. LSID: urn:lsid:zoobank.org:pub:0362DDF4-D56E-4431-9639-3DBF772E221C. Publication LSID: urn:lsid:zoobank.org:pub:0362DDF4-D56E-4431-9639-3DBF772E221C
Figures 2A and 5–10.

*Material examined*: 5 specimens from Nosy Be 13°29′S, 48°21′E; 3 specimens from Morondava, Madagascar 20°17′S, 44°17′E.

Type and paratypes – Nosy Be, Madagascar 13°29′S, 48°21′E.

**Table 1 Pairwise values of Cytochrome Oxyginase Subunit I (COI) among different species of *Chthamalus*.** The analysis involved 17 nucleotide sequences of Chthamalus. Analyses were conducted using the Maximum Composite Likelihood model (*Tamura, Nei & Kumar, 2004*). A total of 458 positions were included in the final dataset. Analyses were conducted in MEGA7 (*Kumar, Stecher & Tamura, 2016*).

| | C. barilani | C. williamsi | C. fragilis | C. alani | C. dentatus | C. challengeri | C. newmani | C. bisinuatus | C. mexicanus | C. dalli | C. angustitergum | C. hedgecocki | C. proteus | C. malayensis | C. cortezianus | C. montagui | C. stellatus |
|---|---|---|---|---|---|---|---|---|---|---|---|---|---|---|---|---|---|
| C. barilani | | | | | | | | | | | | | | | | | |
| C. williamsi | 0.175 | | | | | | | | | | | | | | | | |
| C. fragilis | 0.174 | 0.190 | | | | | | | | | | | | | | | |
| C. alani | 0.169 | 0.172 | 0.174 | | | | | | | | | | | | | | |
| C. dentatus | 0.180 | 0.185 | 0.176 | 0.189 | | | | | | | | | | | | | |
| C. challengeri | 0.172 | 0.183 | 0.172 | 0.162 | 0.161 | | | | | | | | | | | | |
| C. newmani | 0.185 | 0.158 | 0.182 | 0.035 | 0.197 | 0.165 | | | | | | | | | | | |
| C. bisinuatus | 0.194 | 0.172 | 0.220 | 0.215 | 0.196 | 0.168 | 0.207 | | | | | | | | | | |
| C. mexicanus | 0.177 | 0.178 | 0.190 | 0.087 | 0.204 | 0.163 | 0.082 | 0.203 | | | | | | | | | |
| C. dalli | 0.189 | 0.169 | 0.199 | 0.178 | 0.199 | 0.075 | 0.181 | 0.182 | 0.173 | | | | | | | | |
| C. angustitergum | 0.181 | 0.205 | 0.224 | 0.178 | 0.184 | 0.194 | 0.170 | 0.203 | 0.187 | 0.180 | | | | | | | |
| C. hedgecocki | 0.175 | 0.175 | 0.190 | 0.090 | 0.198 | 0.168 | 0.085 | 0.200 | 0.036 | 0.170 | 0.182 | | | | | | |
| C. proteus | 0.228 | 0.170 | 0.213 | 0.191 | 0.210 | 0.179 | 0.182 | 0.212 | 0.189 | 0.177 | 0.219 | 0.189 | | | | | |
| C. malayensis | 0.155 | 0.160 | 0.180 | 0.174 | 0.166 | 0.187 | 0.174 | 0.183 | 0.169 | 0.198 | 0.201 | 0.175 | 0.176 | | | | |
| C. cortezianus | 0.180 | 0.188 | 0.201 | 0.170 | 0.171 | 0.190 | 0.178 | 0.194 | 0.184 | 0.209 | 0.190 | 0.192 | 0.184 | 0.160 | | | |
| C. montagui | 0.174 | 0.190 | 0.208 | 0.199 | 0.188 | 0.151 | 0.196 | 0.222 | 0.189 | 0.192 | 0.186 | 0.191 | 0.207 | 0.188 | 0.177 | | |
| C. stellatus | 0.187 | 0.170 | 0.185 | 0.209 | 0.166 | 0.173 | 0.206 | 0.176 | 0.214 | 0.187 | 0.210 | 0.208 | 0.195 | 0.182 | 0.199 | 0.153 | |

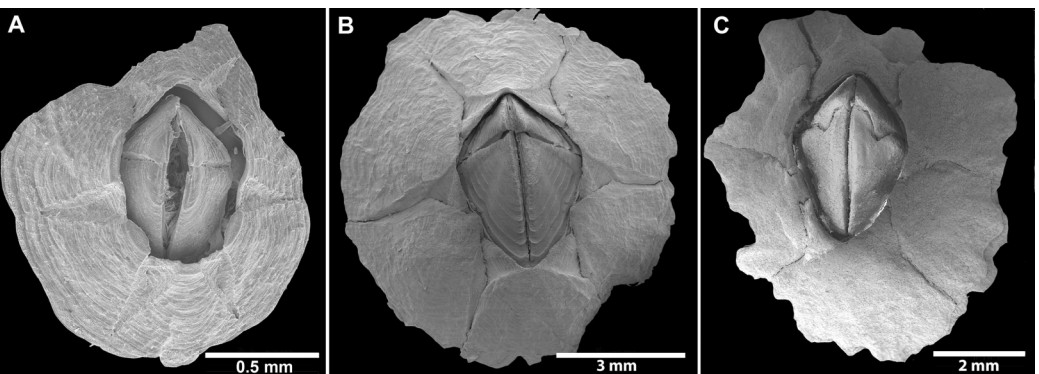

**Figure 5** *Chthamalus barilani* **n.sp. SEM, external view.** *Chthamalus barilani* n.sp. SEM, external view. (A) Newly stalled specimen, Morondava, Madagascar. (B) Non-eroded specimen Nosy-Be, Madagascar. (C) Eroded specimen Belo Sur Mer, Madagascar.     

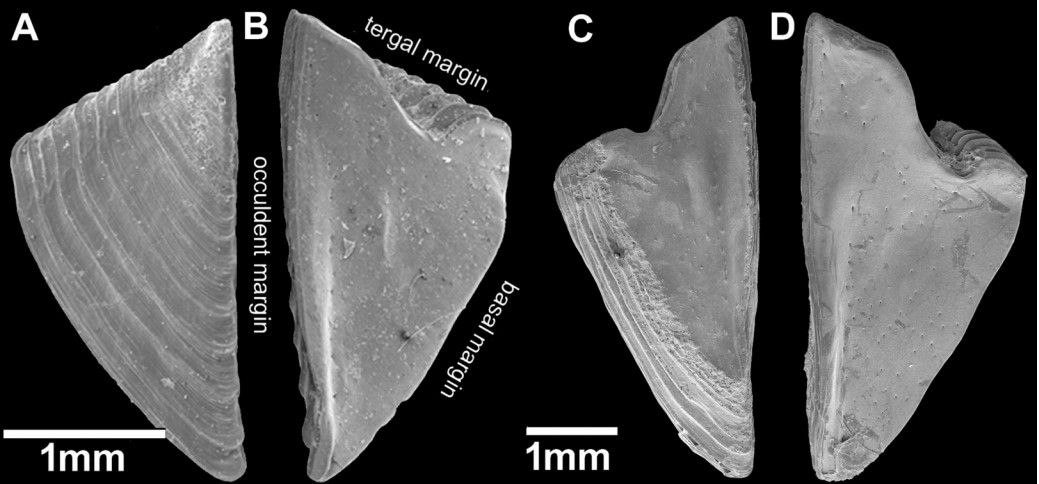

**Figure 6** *Chthamalus barilani* **n.sp. Opercular valves: Scuta.** *Chthamalus barilani* n.sp. Scuta. (A) Non-eroded specimen, outer view. (B) Non-eroded specimen inner view. (C) Eroded specimen outer view. (D) Eroded specimen inner view.     

*Holotype:* HUJICRUSCIRR225 wet sample non eroded specimen

*Paratypes:* HUJICRUSCIRR234 wet sample eroded specimen; HUJICRUSCIRR226 dry sample non eroded specimen; HUJICRUSCIRR227 dry sample eroded specimen

   Diagnosis: Shell pink or dull gray; orifice kite shape; parietes with shallow radial ribs; basis membranous, scutum triangular, concentric growth lines in non-eroded specimens. Tergum nearly rectangular, articular ridges not developed, spur short rounded, basal margin straight. Conical spines on posterior margins of anterior ramus of cirrus I. Bidenticulate setae without basal guards on terminal segments of cirrus II. Penis without basidorsal point.

   Description: Shell (Figs. 2A, 2B and 5A–5C) low conical, carino rostral diameter up to 2 cm. In young non-eroded specimen, the shell is pink with spaced radiating low ridges (Figs. 2B, 5A and 5B). In eroded specimen the radiating ridges are worn and not distinctive

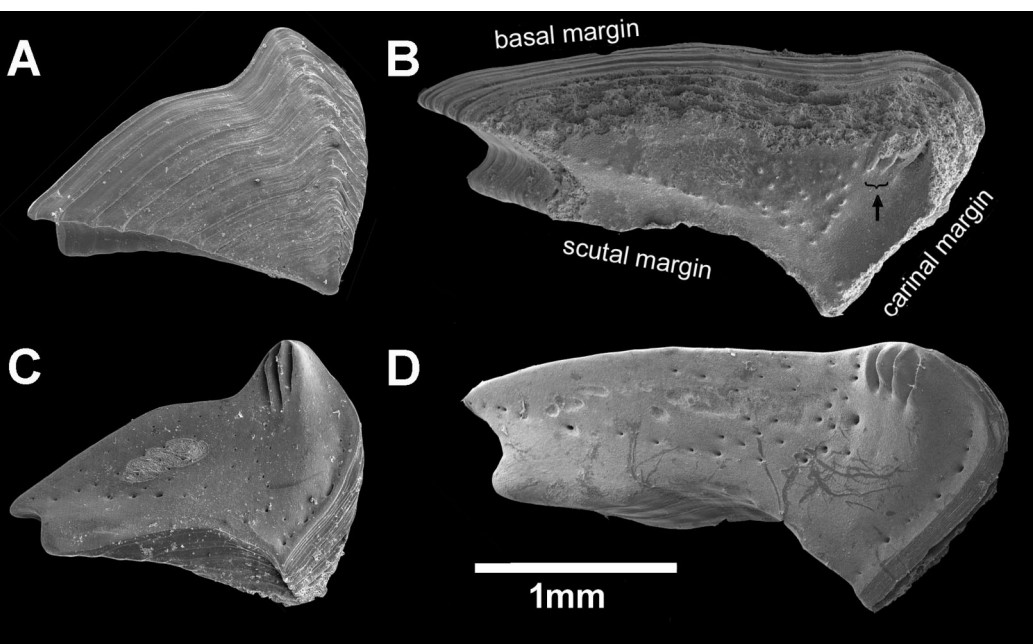

**Figure 7** *Chthamalus barilani* **n.sp. Opercular valves: Terga.** *Chthamalus barilani* n.sp. Terga. (A) Non-eroded specimen outer view. (B) Eroded specimen outer view. (C) Non-eroded specimen inner view. (D) Eroded specimen outer view, arrow indicates exposed depressor crests.

(Figs. 2A and 5C). Opercular opening orifice kite shape, carino-rostral diameter of opercula about half total diameter. In non-eroded specimens, the joint between the paired scuta and terga forms the shape of an arrow head (Figs. 2B and 5B), in eroded specimens, the joint between terga and scuta is sinusoidal (Figs. 2A and 5C).

Scutum (Figs. 6A–6D): triangular, growth lines on outer surface in non-eroded specimens (Figs. 6A and 6B); in eroded specimens worn growth lines. Angle between tergal margin and occludent margins less than 90°. Tergal margins, articular furrow curved in, shallow V shape in non-eroded (Fig. 6), deep V shape in eroded (Fig. 6). Pits for adductor and lateral adductor muscle very shallow or not conspicuous, no adductor ridge. Small holes for muscles attachment on inner surface. In eroded specimens, the occludent margins (Figs. 6C and 6D) are relatively longer than in non-eroded specimens, and the articular furrow is deeper.

Tergum (Figs. 7A–7D): elongated, higher than wide, growth lines on outer surface, shallow groove between basal margin and angle of carinal and scutal margin, carinal margin curved. Inner surface two to three crests for depressor muscle at upper angle of carinal margin. Small pits for muscles on internal surface. Scutal margin nearly straight, low short. articular ridge.

Labrum (Fig. 8A): bilobed, wide V shape notch fringed with fine hairs and small teeth. Mandibular Palpus (Fig. 8B) elongated, leaf like, long simple setae at the distal part, short on the upper margin, spines along lower margins. Mandible (Fig. 8): quadridentate, three big teeth occupy 4/5 of cutting edge, fourth tooth small bident. Comb short lower edge of surface of mandible close to cutting edge with short, simple setae. One big and two

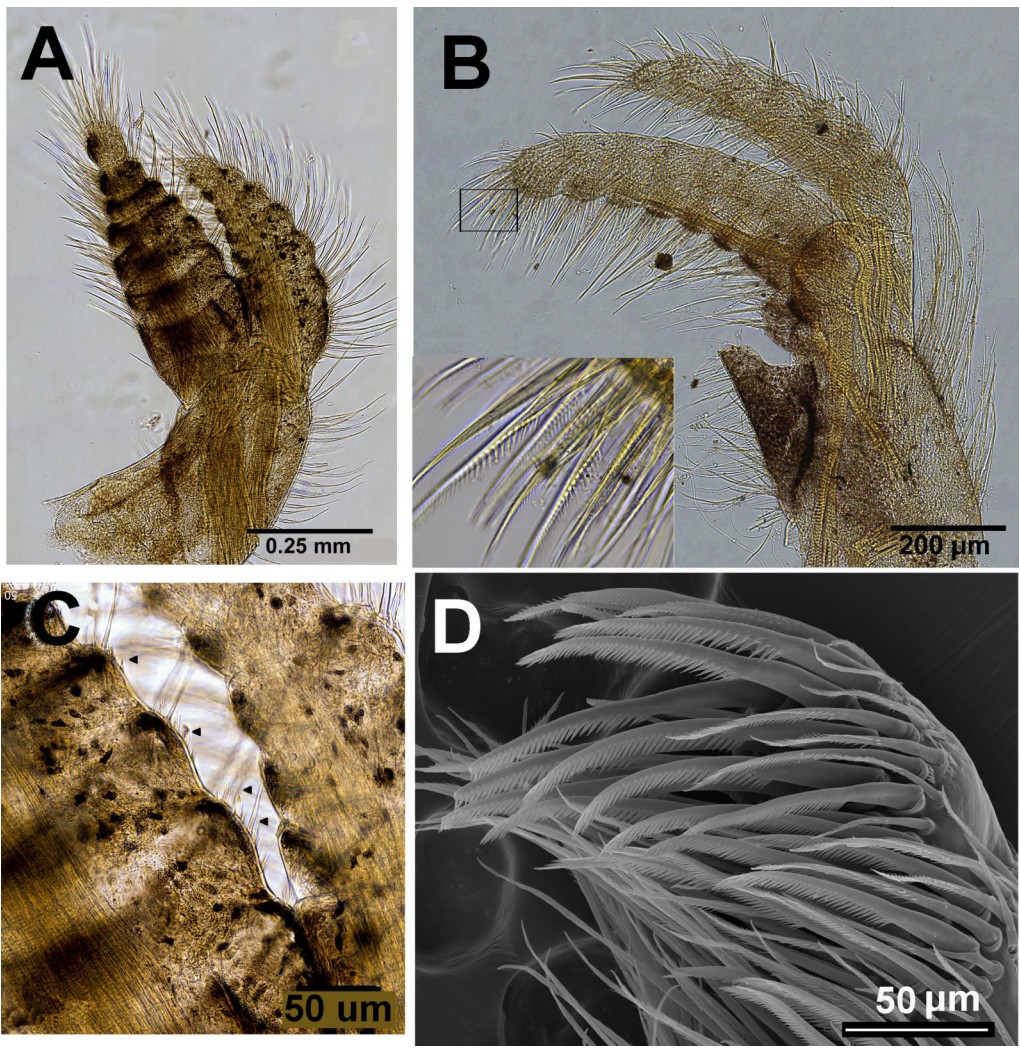

**Figure 8 *Chthamalus barilani* n.sp. first and second cirri.** *Chthamalus barilani* n.sp. Cirri I–II. (A) Cirrus I. (B) Cirrus II inset setae on terminal segments of posterior ramus. (C) Cirrus I conical spines on anterior ramus indicated. (D) SEM of bidenticulate setae without basal guards on terminal segment.

smaller spines on lower angle. Maxilla (Figs. 8C and 8D): bilobed, concave shallow notch without setae divides the cutting margin. Simple setae along anterior margin and distal part. Maxillule (Fig. 8F): cutting edge divided by a notch, upper part two stout spines 5–6 shorted spines, few short setae in notch. Below notch 8–9 stout spines, fine seta at lower part of cutting edge. Simple setae on upper and lower margins. Surface of maxillule close to cutting edge with short, simple type setae.

Cirri (terminology of setae according to *Pitombo (1999)*)

Cirrus I (Figs. 9A–9C) rami unequal, anterior ramus 7–8 segments, posterior shorter 5-6 segments, both rami with simple and denticulated setae. Conical spines on posterior margins of anterior ramus (Fig. 9C). Cirrus II (Figs. 9B and 9D) rami unequal, anterior ramus 6 to 8 segments, posterior ramus 4 to 7-segments, the two distal segments bear bidenticulate setae without basal guards (Figs. 9B inset, 9D). Simple and serrulate setae on

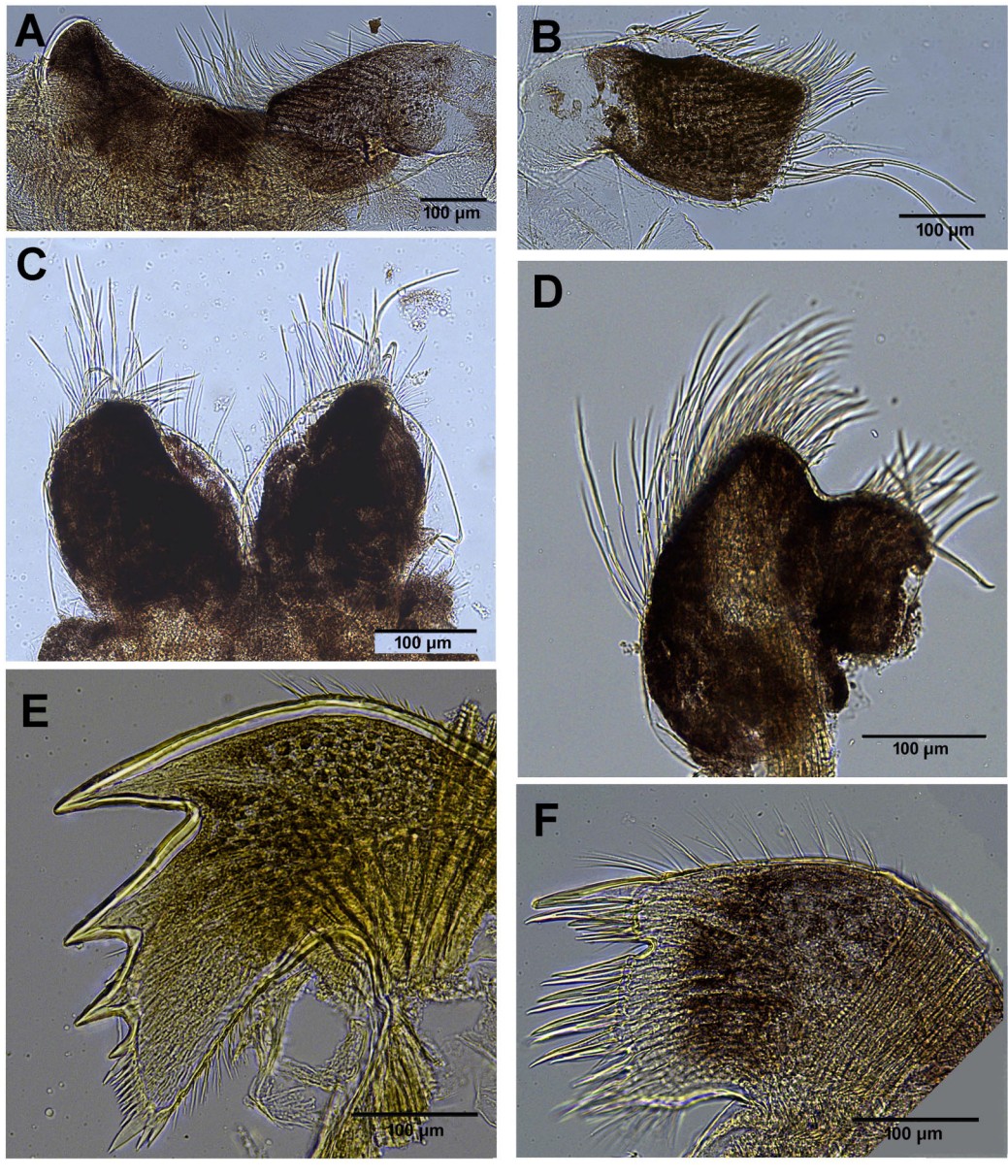

**Figure 9** ***Chthamalus barilani* n.sp. mouth parts.** *Chthamalus barilani* n.sp. mouth parts. (A) Labrum. (B) Pair of maxillae. (C) Maxillule. (D) Mandibular palp. (E) Mandible. (F) Maxilla.

all segments of both rami. Cirri III–VI (Figs. 10A–10D) arrangement of setae on cirri similar. Cirrus III (Fig. 10A) anterior and posterior rami similar in length, 12 segments on each, face of intermediate segments of both rami bear three to four pairs of simple setae, distal setae of segment are longest proximal are shortest. Back of rami at joints between segments, two to three short setae. Terminal segment with three long simple setae. Cirrus IV (Fig. 10B) 14 segments on both rami. Cirrus V (Fig. 10C) 16 segments on both rami. Cirrus VI (Fig. 10D) the longest, 16 segments on anterior ramus, 17 on posterior

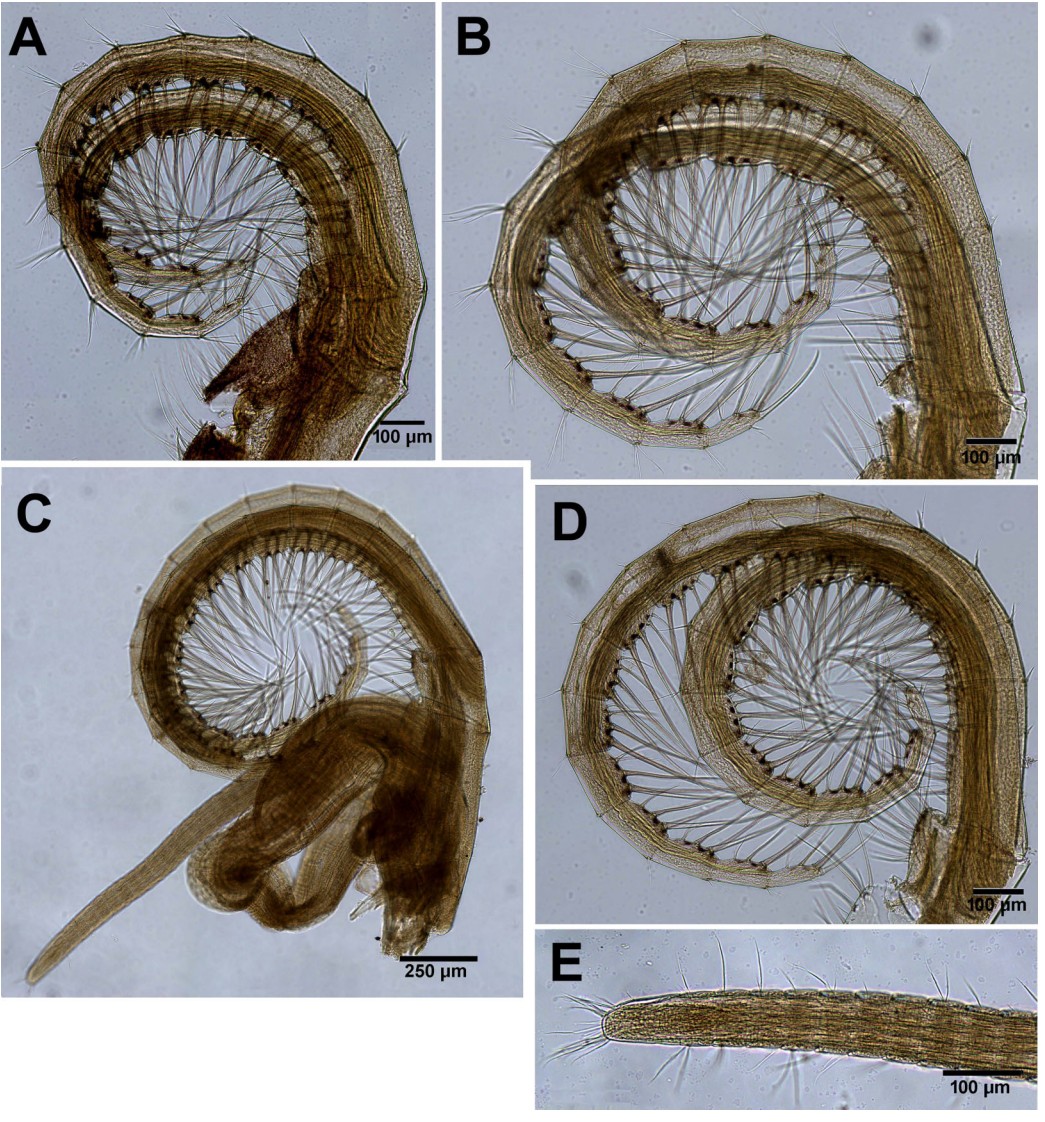

**Figure 10** ***Chthamalus barilani*** **n.sp. Cirri III–VI.** *Chthamalus barilani* n.sp. Cirri III–VI. (A) Cirrus III. (B) Cirrus IV. (C) Cirrus VI and penis. (D) Cirrus V. (E) Distal end of penis.

ramus. Penis (Fig. 10E) long finely annulated, gradually tapering, setae scattered along penis tip with fine setae, no basidorsal point.

*Etymology*: In appreciation of Bar Ilan University, Israel, the host and supporter of our research for more than four decades.

*Remarks*: One of the remarkable features of *C. barilani* is the variability in external morphology particularly in the opercular valves, seen with age or exposure to erosion. This may lead to their allocation to different species. In the newly settled specimen, about 1 mm of the rostro-carinal diameter of the suture between the opercular valves is cross shaped (Fig. 5A) but in the bigger, non-eroded, specimens the suture between the valves is

shaped like an arrowhead (Fig. 5B), while in older eroded specimens, it is sinusoidal (Figs. 5C, 6C–6D). In the eroded specimens the surface is worn and the growth lines are not noticeable (Fig. 6C). The occludent margins are longer relatively to the non-eroded specimens. As already described (Foster, 1974; Achituv & Safriel, 1980), variability of the exterior of barnacles as a result of erosion is an accepted finding. In *C. barilani*, besides erosion, differential growth also influences morphological variability. This becomes apparent when examining the inner side of the opercular valves, which is not affected by erosion, and the position of the tergal depressor crests. The crests, which in non-eroded young specimens (Fig. 7C) are located in the angle between the basal and carinal margins, "shift" towards the middle of the carinal margin in bigger eroded specimens (Fig. 7D). This differential growth pattern affects the morphology of the opercula, and thereby influences the results of erosion on the external surface of shell and opercula; in highly eroded specimens the depressor bases of the crests are exposed (Fig. 7B, arrow).

There is a large overlap in the geographical distributions of *C. barilani* and an unrecognized, un-named evolutionarily significant unit (ESU) *Tetraclita* sp. nov. reported by *Tsang et al. (2012b)*. These authors suggested that the distribution and connectivity of this ESU, as well as of other species of *Tetraclita*, are determined by the circulation systems of the WIO, which in the case of *Tetraclita* sp. nov. is the East Africa Coastal current (EACC) (*Swallow, Schott & Fieux, 1991*). We suggest that the same oceanographic system may also control the geographic distribution of *C. barilani*.

### *Chthamalus malayensis* Pilsbry, 1916.

Pilsbry, 1916 (310–311; pl. 72 6-6B).

Morphological examination of specimens of *Chthamalus* from Zanzibar and Tanzania (Dar es Salaam) reveals that the dominant species of barnacles on rocky shores belongs to *C. malayensis* since the shape of the shell matches the description of *C. malayensis* by Pilsbry (1916; Plate 72: 5, 5A) and *Tsang et al. (2012a)*. The diagnostic characteristics of the *C. malayensis* group are conical spines on the dorsal surface of the anterior ramus of cirrus I and bidenticulated setae with a basal guard on the terminal segments of cirrus II (*Southward & Newman, 2003*). These structures were found on specimens of *Chthamalus* from Tanzania and Zanzibar (Figs. 11C–11E) and clearly indicate that *Chthamalus* from these localities belong to the nominal species *C. malayensis*. However, as already described, the COI and NaKA data revealed that *C. malayensis* is not the only species of *Chthamalus* found in Tanzania and Zanzibar. Sequences obtained from several specimens of *Chthamalus* from these regions clustered with the group of *Chthamalus* from Madagascar, indicating that the populations of these two regions comprise two sympatric species of *Chthamalus*.

Examination of cirri in the specimens from Tanzania and Zanzibar reveals differences between them and those of specimens from the IWP. One of the differences is the presence of setae with triple guards on the terminal segments of cirrus II (Fig. 11). However, *Tsang et al. (2012a)* concluded that since these features are common to a number of clades, they are not suitable for taxonomic identification.

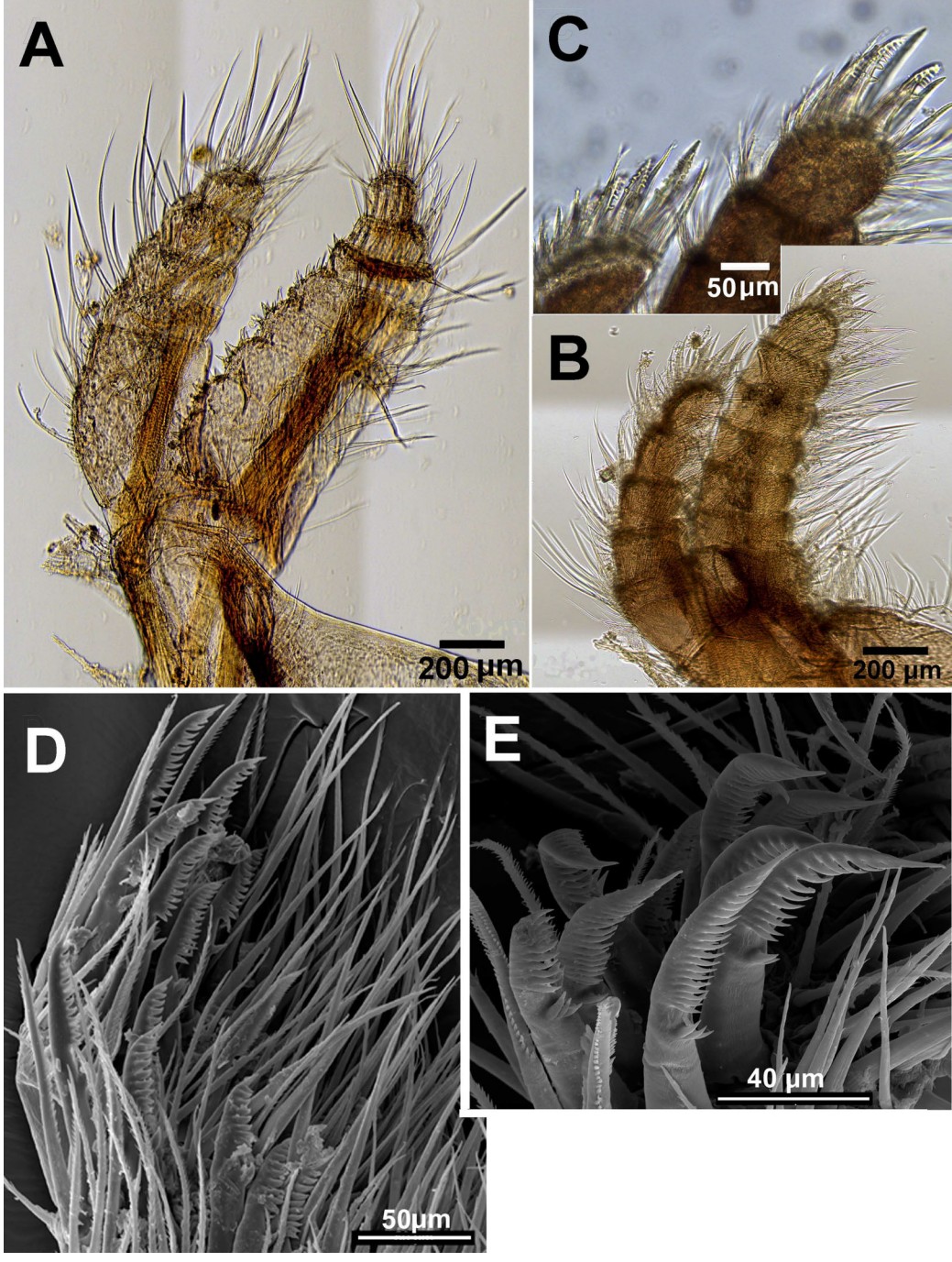

**Figure 11 *Chthamalus malayensis* from Dar E Salaam, Tanzania.** *Chthamalus malayensis* from Dar E Salaam, Tanzania. Cirii I–II. (A) Cirrus I conical spines on. (B) Cirrus II. (C) Cirrus II terminal segments. (D) SEM of simple and bidenticulate setae with basal guards on terminal segment of cirrus II. (E) Bidenticulated setae with basal guards.

*Tsang et al. (2008, 2012a)* used COI sequence data to demonstrate the presence of four clades within the IWP populations of *C. malayensis*. In addition to the molecular differences, these clades also have distinguishing arthropodal characteristics

(*Tsang et al., 2012a*, Figs. 4E–4F). The main differences are in the number of segments with conical spines on the anterior rami of cirrus I and of cirrus II, the number of setae on the terminal segments of cirrus II, and the number of setules in the setae. The authors argue that these four clades are actually four distinct biological species, but since there are no diagnostic morphological characters that can separate the four species with confidence, they should therefore be regarded as "cryptic species". Based on the molecular analyses and the morphology of setae, we similarly conclude that the East African specimen is another cryptic species of the *C. malayensis* cluster. The pairwise distances within the cluster of specimens from Tanzania and between this and representatives from the four clades identified as cryptic species were calculated and presented in Table 2. Distances within locations ($n = 40$, mean = 0.0091) and distances between locations ($n = 191$, mean = 0.1169) the two means were compared. Since distances are non-independent, the comparison was performed by boot-strapping, by creating 1,000 random samples. Not surprising, the mean distance between clusters was significantly larger than the mean distance within clusters (estimated $p < 0.001$). This support the morphological observations. Cryptic species of barnacles are known from other species of *Chthamalus*, for example *C. moro* (*Wu et al., 2014*) and *C. challengeri* (*Cheang et al., 2015*), as well as in other genera, namely *Tetraclita* (*Chan, Tsang & Chu, 2007*) and *Hexechamaesipho* (*Tsang et al., 2013*).

The two COI sequences of *C. malayensis* from Queensland form a clade that clusters with the Taiwan (TW) clade. The bootstrap support of the node that separates this clade from the TW clade is 0.99, the pairwise distances are low (0.024–0.032), and it is tempting to assume that *C. malayensis* from Queensland is part of the TW clade. However, the geographic distance between Queensland and Taiwan, the source of the TW clade, and the observation that the SC clade is located in the Philippines between the two, does not accord with an allopatric model. Moreover, the low number of sequences from Queensland does not allow us to conclude the existence of an additional clade.

The distribution of the four previously described clades of *C. malayensis* is separated from that of the East African clade by the Arabian Sea, and part of the gap is occupied by another species, *C. barnesi*, which has been recorded in the Gulf of Oman and the coast of Yemen (*Shahdadi & Sari, 2011*). It is tempting to hypnotize that propagules of *C. malayensis* are carried by the North East Monsoon current across the Arabian sea from the Eastern coast of India to the western coast of Africa. This hypothesis could be supported by dispersal ability of larvae across the vast distance between the two coasts. Larval development of the six naupliar stages, from hatching to the non feeding cypris stage, of *C. malayensis* from Hong Kong took 20 days at 2 °C and 14 days at 28 °C (*Yan & Chan, 2001*). For the same species from Mumbai, India *Karande & Thomas (1976)* reported that larval development took between 7 to 12 days, these authors did not indicate the temperature of their culture. This period does not allow the traverse of the planktonic larvae across the Arabian Sea. However, the existence of islands inhabited by *C. malayensis* that serve as stepping stones across the Arabian Sea can facilitated the passage of this sea. Due to a lack of information on the distribution of *Chthamalus* in the Arabian Sea this remains unproved speculation. It is more likely that the vicariance model

**Table 2  Pairwise distance values of Cytochrome Oxyginase Subunit I (COI) within and among different clades of *Chthamalus malayensis*.**

| | IM 1 | IM 2 | IM 3 | IM 4 | IM 5 | TW 1 | TW 2 | Tw 3 | Tw 4 | SC 1 | SC 2 | SC 4 | SC 3 | CI 2 | CI 1 | CI 3 | DeS 1 | Des 6 | DeS 2 | DeS 5 | DeS 3 |
|---|---|---|---|---|---|---|---|---|---|---|---|---|---|---|---|---|---|---|---|---|---|
| IM-Clade 1 | | | | | | | | | | | | | | | | | | | | | |
| IM-Clade 2 | 0.011 | | | | | | | | | | | | | | | | | | | | |
| IM-Clade 3 | 0.006 | 0.006 | | | | | | | | | | | | | | | | | | | |
| IM-Clade 4 | 0.009 | 0.009 | 0.004 | | | | | | | | | | | | | | | | | | |
| IM-Clade 5 | 0.007 | 0.007 | 0.002 | 0.006 | | | | | | | | | | | | | | | | | |
| TW-clade1 | 0.093 | 0.093 | 0.095 | 0.096 | 0.098 | | | | | | | | | | | | | | | | |
| TW-clade 2 | 0.086 | 0.086 | 0.088 | 0.089 | 0.091 | 0.011 | | | | | | | | | | | | | | | |
| TW-clade 3 | 0.086 | 0.086 | 0.089 | 0.084 | 0.091 | 0.015 | 0.008 | | | | | | | | | | | | | | |
| TW-clade4 | 0.090 | 0.091 | 0.093 | 0.093 | 0.095 | 0.011 | 0.007 | 0.011 | | | | | | | | | | | | | |
| SC-clade 1 | 0.157 | 0.152 | 0.160 | 0.161 | 0.163 | 0.142 | 0.137 | 0.149 | 0.145 | | | | | | | | | | | | |
| SC-clade 2 | 0.129 | 0.127 | 0.132 | 0.132 | 0.134 | 0.121 | 0.121 | 0.127 | 0.123 | 0.027 | | | | | | | | | | | |
| SC-clade 4 | 0.139 | 0.137 | 0.142 | 0.143 | 0.145 | 0.128 | 0.128 | 0.134 | 0.130 | 0.025 | 0.013 | | | | | | | | | | |
| SC-clade 3 | 0.140 | 0.137 | 0.143 | 0.143 | 0.145 | 0.123 | 0.123 | 0.129 | 0.126 | 0.021 | 0.009 | 0.007 | | | | | | | | | |
| CI-clad 2 | 0.142 | 0.142 | 0.144 | 0.145 | 0.147 | 0.130 | 0.130 | 0.136 | 0.132 | 0.082 | 0.069 | 0.071 | 0.071 | | | | | | | | |
| CI-clade 1 | 0.144 | 0.145 | 0.147 | 0.148 | 0.150 | 0.130 | 0.130 | 0.136 | 0.132 | 0.082 | 0.069 | 0.075 | 0.071 | 0.004 | | | | | | | |
| CI-clade 3 | 0.144 | 0.145 | 0.147 | 0.148 | 0.150 | 0.133 | 0.133 | 0.139 | 0.135 | 0.082 | 0.071 | 0.071 | 0.071 | 0.004 | 0.007 | | | | | | |
| Dar e Salaam 1 | 0.086 | 0.086 | 0.088 | 0.089 | 0.088 | 0.081 | 0.075 | 0.079 | 0.079 | 0.157 | 0.141 | 0.149 | 0.146 | 0.139 | 0.139 | 0.141 | | | | | |
| Dar e Salaam 6 | 0.091 | 0.091 | 0.093 | 0.094 | 0.093 | 0.086 | 0.079 | 0.084 | 0.084 | 0.163 | 0.146 | 0.154 | 0.152 | 0.134 | 0.134 | 0.137 | 0.006 | | | | |
| Dar e Salaam 2 | 0.088 | 0.088 | 0.091 | 0.091 | 0.091 | 0.084 | 0.077 | 0.082 | 0.081 | 0.160 | 0.143 | 0.151 | 0.149 | 0.141 | 0.141 | 0.144 | 0.004 | 0.009 | | | |
| Dar e Salaam 5 | 0.091 | 0.091 | 0.093 | 0.093 | 0.093 | 0.086 | 0.079 | 0.084 | 0.083 | 0.163 | 0.146 | 0.154 | 0.152 | 0.139 | 0.139 | 0.141 | 0.004 | 0.006 | 0.004 | | |
| Dar e Salaam 3 | 0.088 | 0.088 | 0.091 | 0.091 | 0.091 | 0.084 | 0.077 | 0.082 | 0.081 | 0.160 | 0.144 | 0.152 | 0.149 | 0.141 | 0.141 | 0.144 | 0.006 | 0.011 | 0.006 | 0.006 | |
| Dar e Salaam 4 | 0.093 | 0.093 | 0.095 | 0.096 | 0.095 | 0.088 | 0.081 | 0.086 | 0.081 | 0.166 | 0.149 | 0.157 | 0.155 | 0.149 | 0.149 | 0.152 | 0.009 | 0.015 | 0.009 | 0.009 | 0.011 |

**Note:**

Pairwise distance values of Cytochrome Oxyginase Sub unit I (COI) within and among different clades of *Chthamalus malayensis* from the Indo-West Pacific (*Tsang et al., 2012a*) and East African clades. Analysis was conducted in MEGA7 (*Kumar, Stecher & Tamura, 2016*). Abbreviation: IM Indo-Malayan; TW Taiwan; SC South China Sea: CI Christmas Island.

explains the present-day clade distribution. *Tsang et al. (2008)* suggested that the Indo-Malay clade population of *C. malayensis* attained its present range by postglacial recolonization from the Pacific, followed by transport of planktonic larvae westward from the Pacific through to the central and West Indian Ocean and its adjacent seas. The subsequent founder effect during re-colonization led to a reduction in genetic diversity.

Nevertheless, it is possible that the glacial cycles during the Pleistocene broke the once continuously distributed species into geographically separated populations. This separation shaped genetic variation across the geographical ranges of many taxa. Cooling and increased ice cover and consequent low sea level, isolated populations into refugia. Once the glaciers retreated, population expansion from these refugia resulted in genetic differences in the populations. *Dolby et al. (2016)* applied the refugia model to the reestablishment of estuarine habitats along the eastern Pacific coast between California and Baja California.

### *Chthamalus barnesi* Achituv & Safriel, 1980
*Jones, 1986*, p. 145, pl. 39; *Shahdadi & Sari, 2011*, p. 747, Fig. 2.

*Broch (1927)* was the first that to identify *Chthamalus* in the Red Sea, suggesting that it belongs to *C. challengeri* and resembles the subspecies *C. nipponensis*. *Stubbings (1961)*, in his description and report on cirripedes of the West Indian Ocean, listed *C. malayensis* as the species of *Chthamalus* inhabiting this oceanic area. *Achituv & Safriel (1980)*, who collected samples in the Red Sea, recognized *Chthamalus* from this area as a distinct species, *C. barnesi*. Even after this description of *C. barnesi*, the taxonomic position of *Chthamalus* from the West Indian Ocean remained unclear. *Jones (1986)* considered the population of *Chthamalus* from the shore of Kuwait and the Arabian Gulf as *C. malayensis*. However, according to *Shahdadi & Sari (2011)* no specimens of *C. malayensis* were observed in 23 localities along the Persian Gulf's shores, from Gwater Bay at the border of Pakistan to Arvand Estuary at the border of Iraq.

*Southward & Newman (2003)* considered the taxonomic status of *Chthamalus* spp. from the Red Sea, the Persian Gulf, Somali, Kenya, and Pakistan to be unclear. They allocated them to the "*challengeri*" subgroup and named them *C. cf. challengeri*. However, the close resemblance of outer parts of the specimens from the Red Sea to other species led *Southward & Newman (2003)* to suggest that the West Indian population belongs to a different species, close to *C. challengeri*. It is the difference in the inner morphology of scuta, trophi, and cirral appendages that led *Achituv & Safriel (1980)* to conclude that the population of the Red Sea does not belong to *C. challengeri* but to a different species. They attributed the variability in shell morphology to age and different degrees of erosion of the shell. Notably, variation in shell color may result from different algae infecting the shells.

Our molecular analyses confirm the conclusions based on the morphological features that the Red Sea population is a distinct clade within the other Indo-West Pacific species of *Chthamalus*. The COI and NaKA sequences obtained from the Red Sea specimen cluster with those of the specimens from Yemen, the Arabian Sea, the Gulf of Oman, and the Persian Gulf, confirming the finding of *Shahdadi & Sari (2011)*.

### *Chthamalus dentatus* Krauss, 1848
*Darwin, 1854*; *Barnard, 1924*; *Stubbings, 1967*

*Chthamalus dentatus* is one of the species of *Chthamalus* that is easily recognized by distinct external features. As the described by *Darwin (1854)*: "the sutures formed by interlocking teeth". The toothed edges of the shell plate are clearly visible in specimens collected in Morondava in Madagascar. We could not find this species at Nosy Be. We have no information on its northern boundary on the African continent's eastern coast; it was not found in Dar es Salaam in Tanzania, where the *C. malayensis* is found.

The pairwise distance of specimens from Madagascar and specimens from Durban and Arniston in South Africa ranges from 0.004 to 0.017, which is the range found within species of *Chthamalus*. We therefore conclude that the Madagascar's *C. dentatus* population is identical to those found along the southern and eastern coasts of Africa.

## DISCUSSION

In his recent survey *Wares (2020)* reviewed how molecular methods had contributed to our knowledge and understanding of aspects in the biology of the "Small, flat, gray and cryptic" barnacle *Chthamalus*. Species of *Chthamalus* are phenotypically similar and shell and opercular are poor criteria for classifying these barnacles (*Wares et al., 2009*; *Pérez-Losada et al., 2012*). The use of molecular tools promoted the description of new species, the understanding of the phylogenetic genetic diversity between and within species and the biogeography of the genus. In the present study we combined morphological characteristics and molecular markers to identify *Chthamalus* species and their phylogeography in the West West Indian Ocean.

*Dando & Southward (1980)* recognized four distinct informal groups of *Chthamalus* based on morphological characters, namely, the 'challengeri', 'fissus', 'stellatus', and 'malayensis' groups. They suggested that these informal groups represent monophyletic entities, supported by the known distributions of the species within this genus. This information was summarized by *Southward & Newman (2003)* and subsequently reviewed by *O'Riordan, Power & Myers (2010)* who compiled a table listing the recognized species, and relating them to their geographical distribution. They presented maps showing the geographical distribution of these groups.

The morphological characteristics used to distinguish between the groups are the presence and absence of conical spines on the cirrus I and the morphology of setae on the terminal segments of cirrus II (*Southward & Newman, 2003*). According to these criteria, *Chthamalus barilani*, the species described in the present study, belongs to the 'stellatus' group characterized by the presence of conical spines on cirrus I and complex setae with no basal guards on cirrus II. *C. barnesi*, which is also found in the WIO, although not included in the table compiled by *O'Riordan, Power & Myers (2010)*, lacks conical spines on cirrus I and has no basal guards on the complex setae of cirrus II (*Shahdadi & Sari, 2011*), and thus should be included in the 'challengeri' group.

We used COI sequences of the genus *Chthamalus* to examine whether the morphological grouping within this genus, is supported by the molecular data. In addition to the species included in the table presented by *O'Riordan, Power & Myers (2010)* we

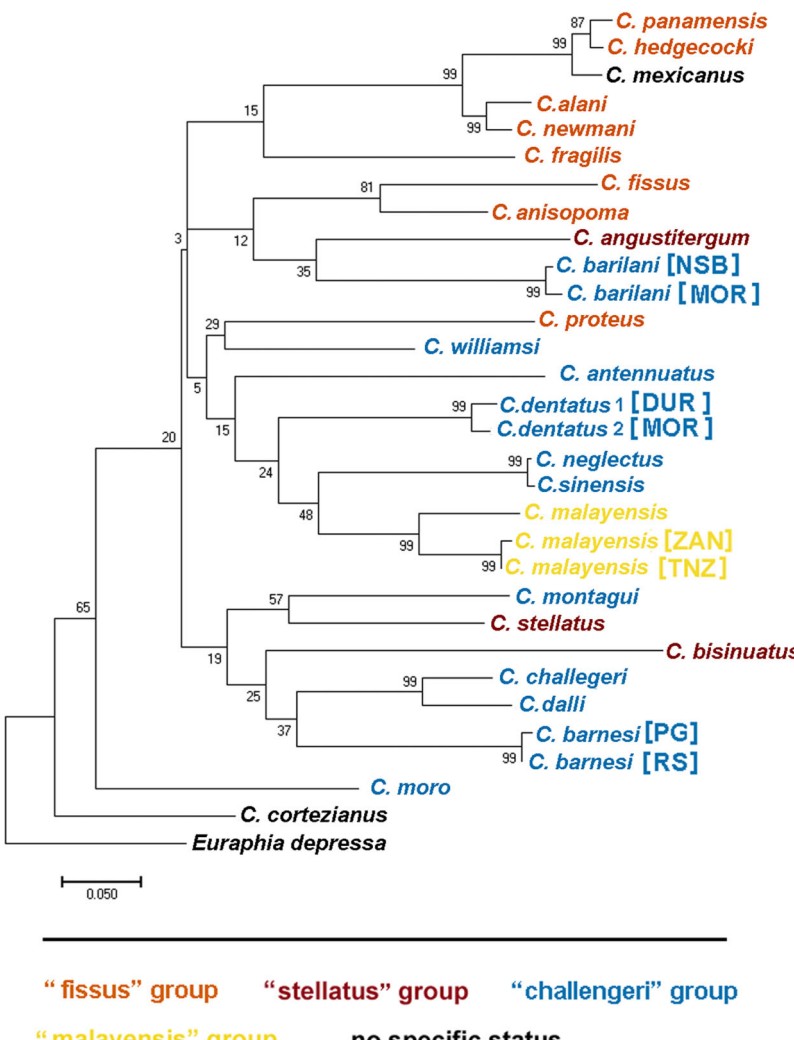

"fissus" group      "stellatus" group      "challengeri" group

"malayensis" group      no specific status

**Figure 12 Maximum likelihood phylogeny based on COI gene sequences of nominal species of *Chthamalus*.** Maximum likelihood phylogeny based on COI gene sequences of informal groups of nominal species of *Chthamalus* based on table 1 in *O'Riordan, Power & Myers (2010)*. Colour indicate groups. Sequences obtained from GeBank (Supplemental Material 1) and generated in the present study, locality of material in square parentheses. Abbreviations of localities: NSB = Nosy-Be, Madagascar; MOR = Morondava, Madagascar ; DUR = Durba, South Africa; ZAN = Zanzibar, Tanzania; TNZ = Dar e Salaam,Tanzania; PG = Persian Gulf, Iran. RS = Red Sea, Israel.

included *Chthamalus williamsi Chan & Cheang, 2015*, which, according to these authors belongs to the 'challengeri' group, together with *Chthamalus alani*, and *Chthamalus newmani* Chan, 2016. The last two species were described by *Chan et al. (2016)*, who did not discuss the presence of conical spines on cirrus I of these species. Examination of the figures accompanying the description does negate such spines, but does indicate the presence of bidenticulated setae with basal guards that characterize the 'fissus' group. For two species, *Chthamalus cortezianus Pitombo & Burton, 2007* and *Chthamalus mexicanus Laguna, 1985*, the original description was insufficient to allow us to assign them a group. *Dando & Southward (1980)* suggested that it is probable that the

morphological characters used to separate groups reflect the presence of a common ancestor and phylogeographic relationships. However, the conformity between the phylogenetic tree (Fig. 12) and the grouping based on morphological parameters is only partial. There are clades that cluster species from different groups; for example, the phylogenetic analyses placed *C. stellatus* and *C. montagui*, that belong to different informal groups, on the same clade. This assembling is also supported by *Shemesh et al. (2009)* and by *Pérez-Losada, Høeg & Crandall (2004*; *Pérez-Losada et al., 2012*, *2014)*. *C. dentatus* and *C. malayensis* that belong to two distinct informal groups are clustered in the same clade, this finding is also supported by *Pérez-Losada et al. (2012*, *2014)*.

Species of the 'fissus' group with basal guards on the complex setae, currently found in the eastern Pacific and Caribbean, could be monophyletic, derived from an amphiamerican ancestor after the closure of the isthmus of Panama. The inclusion of *C. barilani* from Madagascar in the same clade as the American species does not agree with this assumption. *Dando & Southward (1980)* also suggested that the 'challengeri' group (lacking both conical spines and basal guards on the complex setae) could be regarded as derived from a common circumpolar ancestor. The clustering of *C. barnesi* from the West Indian Ocean with the West Pacific *Chthamalus* is in accordance with this hypothesis. However, *C. montagui* and *C. dentatus* representing other species of *Chthamalus* found in the Atlantic, are grouped with the Indo-Pacific 'challengeri' group. The disjunct distribution of components of the same informal group on the two sides of Africa does not support the suggestion that the 'challengeri' group is a monophyletic taxon. Neither can the effect be explained by the dispersal or vicariance models.

The difficulties in accepting the concept that the informal groups are monophyletic units led us to conclude that the morphological traits used to separate the groups are derived characteristics that lead to homoplasy. However, it should be noted that the phylogenetic tree is not robust; because it is based on a single marker, and the bootstrap support of some of the nodes is low. Analyses based on more markers, i.e., morphological, molecular, or both, will be required to confirm which approach (monophyletic or homoplastic) is correct.

## ACKNOWLEDGEMENTS

Dr. Yaakov Langzam, The Electron Microscopy Unit of the Mina and Everard Goodman Faculty of Life Sciences Bar Ilan University, helped with the SEM work. Dr. Sue Frumin, Laboratory of Archaeobotany, The Martin (Szusz) Department of Land of Israel Studies and Archaeology Bar Ilan University helped with the preparation of figures using the Olympus SZX10 dissecting microscope. Finally, we thank Ms. Yael Laure and Ms. Ann Avron for helping in editing and improving this manuscript.

### Funding

This study was supported by grant 574/14 of the Israel Science Foundation (ISF) to Yair Achituv: "Following Darwin: The evolution of the acorn barnacles". The funders had no

role in study design, data collection and analysis, decision to publish, or preparation of the manuscript.

## Grant Disclosures
The following grant information was disclosed by the authors:
Israel Science Foundation (ISF): 574/14.

## Competing Interests
Oren Levy is an Academic Editor for PeerJ.

## Author Contributions

- Noa Simon-Blecher performed the experiments, analyzed the data, prepared figures and/or tables, authored or reviewed drafts of the paper, and approved the final draft.
- Avi Jacob performed the experiments, prepared figures and/or tables, and approved the final draft.
- Oren Levy performed the experiments, authored or reviewed drafts of the paper, and approved the final draft.
- Lior Appelbaum performed the experiments, authored or reviewed drafts of the paper, and approved the final draft.
- Shiran Elbaz-Ifrah conceived and designed the experiments, performed the experiments, analyzed the data, authored or reviewed drafts of the paper, and approved the final draft.
- Yair Achituv conceived and designed the experiments, performed the experiments, analyzed the data, prepared figures and/or tables, authored or reviewed drafts of the paper, and approved the final draft.

## DNA Deposition
The following information was supplied regarding the deposition of DNA sequences:

Cytochrom oxygenase sununit I

New sequences

Chthamalus malayensis

Zanzibar, Turtle-island: MW283019–MW283043

Tanzania Dar e Salaam: MW283044–MW283055

Townsville, Queensland Australia: MW283056–MW283057

Singapore: MW283058–MW283059

Chthamalus barilani

Nosy be, Madagascar: MW283060–MW283072

Tanzania, Dar E Sallam: MW283075–MW283079

Chthamalus barnesi

Elat, Israel: MW283080

Nabeq, Egypt: MW283081–MW283082

Ras-Misela, Egypt: MW283082–MW283090

Gul of Oman: MW283091; MW283092.

Yeman, MW283095: MW283096

Persian Gulf: MW283093; MW283094

Chthamalus dentatus

Morondava, Madagascar: MW283106–MW283116

Durban, South Africa: MW283105.

Chthamalua antenuatus

Sydney, Australia: MW283097–MW283098

Chthamalus moro

New Caledonia: MW283102: MW283103.

Chtamalus neglectus

Hong Kong: MW283099: MW283100.

Taiwan: MW283104

Chthamalus challengeri

Japan: MW283101

Retrieved From GenBank

Used as out group COI

Octomeris angolosa: HQ567461

Octomeris brunnea: MN928635

COI sequnces for Tables 1 and Fig. 12

*Chthamalus stellatus*: KY639396

*Chthamalus montagui*: EU699191

Chthamalus newmani: KU356720

Chthamalus williamsi: KM594056

Chthamalus alani FJ858020

*Chthamalus fragilis*: MG767198

Chthamalus dentatus: MT563422

Chthamalus panamiensis: FJ857982

Chthamalus challangeri: HM136274

*Chthamalus bisinuatus*: FJ845850

Chthamalus mexicanus: AF234805

*Chthamalus angustitergum*: FJ858059

Chthamalus hedgecocki: FJ857991

*Chthamalus proteus*: AY823014

Chthamalus malayensis: EU304446

Chthamalus cortezianus: AF234811

*Chthamalus montagui*: KU682188

*Chthamalus stellatus*: KY639396

Chthamalus neglectus: FJ858080

Euraphia depressa AY428050

*Chthamalus fissus* MG431315

Chthamalus moro MK995385

COI sequnces used for Table 2 Chthamalus malayensis (*Tsang et al., 2012a*)

SC clade: EU304377; EU304378; EU304419; EU304420.

CI Clade: JQ755172; JQ755173; JQ755174; JQ755175.

IM clade: EU304399; EU304400; EU304401; EU304402; JQ754847; JQ754848.

TW clade: JQ755120; JQ755121; JQ754732;.
Naka (Sodium Potassium ATPase) Sequences
Chthamalus malayensis:
Zanzibar MW530572–MW530576
Dar E Salaam; Tanzania MW530577–MW530585
Singapore MW530586
Ko Phi Phi; Thailand MW530587
Queensland MW530588
Chthamalus dentatus
Morondava; Madagascar MW530589–MW530594
Durban; South Africa MW530595–MW530601
Port Alfred; South Africa MW530602
Chthamalus barnesi
Ras Misela Red Sea MW530603–MW530613
Gulf of Oman MW530613–MW530622
Persian Gulf MW530623–MW530642
Yeman MW530643–MW530644
Nabeq Red Sea Egypt MW530645–MW530646
Eilat Israel MW530647
Chthamalus barilani
Zanzibar Manngroves MW530648–MW530658
Morondava Madagascar MW530659–MW530673
Nosy Be; Madagascar MW530674–MW530685
Belo Sur Mer; Madagascar MW530686–MW530687
Chthamalus alani: FJ862736

## Data Availability
    The raw data are available in the Supplemental Files.

## New Species Registration
The following information was supplied regarding the registration of a newly described species:
    Publication LSID: urn:lsid:zoobank.org:pub:0362DDF4-D56E-4431-9639-3DBF772E221C
    *Chthamalus barilani* n.sp. LSID: urn:lsid:zoobank.org:act:11D8D2FA-9DE9-472A-A172-A64DDA792346.

## Supplemental Information
Supplemental information for this article can be found online at http://dx.doi.org/10.7717/peerj.11710#supplemental-information.

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
