# Peer review of "Flatfoot in Africa, the cirripede Chthamalus in the west Indian Ocean"

_PeerJ, doi:10.7717/peerj.11710_

## Round 0.1 · original submission · Minor Revisions

I have now received two thorough reviews of your paper. Both reviewers felt your paper will be a valuable contribution after some adjustments. Both have outlined a number of issues for you to consider. Please note that Reviewer 2 has made additional comments/corrections in an annotated pdf file. Be sure to download and look through this additional file as you work on your revised paper. I concur with these reviews and look forward to your revision.

Reviewer 1 ·

Basic reporting

This paper set out to describe the phylogeography of chthamalid barnacles from the West Indian ocean. The authors use morphological descriptions derived from LM, SEM, and macro images in combination with molecular markers (COI and NaKA). Four species-level clades were found in the WIO and adjacent waters, including one new species, the Madagascar-endemic C. barilani. This species is described in good detail. C. malayensis is found to harbor four genetically separated clades/clusters and it is concluded that these are four putatively cryptic species (or one? See below). The authors morphologically describe, and amplified, the four species clusters from the WIO and argue that their relationships differ from other accounts of these species.

I liked that the authors strived to integrate several data layers and that they also discuss dispersal potential. However, the overall readability and coherence of the manuscript need to be substantially improved. At present, it is really difficult to understand what the authors do and why. I have commented on this in detail below.

Front page and abstract:

Author list: Why are the two first authors having different affiliations? 1 and 2 appear to be the same institution.

“.. are key inhabitants of rocky intertidal shores”: There is no reports on the ecological importance of these barnacles. We may surmise that they must play some role in the ecosystem, but every species plays a role. I don’t support this sentence. “Key” is too broad and not supported by empirical data nor experiments. I suggest writing “wide-spread” or “commonly encountered”

“.. which was found in Madagascar”: Is this new species only found there? In that case, it would be nice to mention that it is a new endemic species.

“C. malayensis comprises a group of four genetically differentiated clades
representing four cryptic species. The newly identified African clade is genetically different from these clades and the pairwise distances between them justify the conclusion that it is an additional cryptic species of C. malayensis”. I am not sure what the authors mean. What is “it is an additional cryptic species”? I thought the authors argued for four cryptic species. Please elaborate or rephrase more clearly.

“This represents the East Northern...”: Northeastern

“merely on morphological data”: Suggest substituting merely with “solely”

“correct taxonomy”: What is correct taxonomy? The taxonomy of species is a continuum and claiming that COI, NaKa and a limited set of morphological characters reveals the universally true taxonomy is probably an oversimplification. How about writing for “…combination of morphology and phylogenetics in barnacle taxonomy”?

Introduction:

L56: I am not convinced that the species are “key inhabitants”. Explain more or provide concrete evidence. Alternatively, write “some of the most commonly encountered intertidal barnacles with a worldwide distribution”.

L57: Darwin wrote four monographs and the one cited is just one of them. It is not a big deal but maybe write “in his monumental monograph on extant acorn barnacles”.

L60: Chthmalinae is currently challenged. Perez-Losada et al. (2012) clearly showed that neither Euraphinae, Chthamalinae, nor Notochthamalinae is monophyletic. Suggest writing either “of the paraphyletic subfamily Chthmalinae” or “of the family Chthmalidae”

L61: Chan et al. (2021; Zoological Journal Linnean Society) recovered 27 species of the genus. Can the authors elaborate on that discrepancy?

L72: East coast?

L91: Suggest adding this statement at the end of the sentence: “…. , although several taxa (from the entire class Thecostraca to subfamilies) are essentially only recognized by molecular analyses (Chan et al. 2021)”. Many barnacle taxa are only monophyletic by molecular analyses.

L94: “of morphological variants different species”. Suggest rewriting or just writing “morphologically different species”

L97: “growth, (mainly opercular),” . Delete comma between growth and ( )

L98: “arthropod appendages” . Suggest writing “segmented”

L99: “distinguish variants, over the past” . Change the comma to a full stop.

L100-101: Why only this reference? Chan et al. 2021 is the most recent and most comprehensive account to date. The studies by Perez-Losada (2001, 2002, 2009, 2012, 2014) should also be cited since they are important contributions.

L103: “of Chthamalid barnacles”. Write chthamalid

L107: “this part of the ocean revealed” . Which ocean? The Ocean in general? Or “these seas”? or “oceans? I am confused.

L116: “Skeletal structures” . What is meant by this statement? The exoskeleton? Or the “shell plates” which are not skeletal by any means.

L129: “according to the manufacturer’s protocol” . This is a very standard way of writing and I am aware that every study using DNA extraction writes it this way. I just don’t like it. Please emphasize that you followed that protocol exactly so that others can repeat the experiment. I can appreciate it if the authors want to keep their statement as it is indeed a standard statement.

L131: “a partial CDS”. Please define CDS

L133-138: “barcode primers LCO1490 and HCO2198 (Folmer et al., 1994) and the primers of Wares et al. (2009)”. Please be more specific. Either use a table or spell out the exact primer sequences, the annealing temperatures, extension temperatures, number of cycles, and volumes used in the PCR reactions including the amount of input DNA. Please also note that you carefully checked (i.e., translated) the obtained COI sequenced. Co-amplifying Nuclear mitochondrial pseudogenes lead to spurious species assignments. It is not universally true that NUMTs possess truncated protein structures or stop codons, but still, it is usually a pretty good indicator that a NUMT was co-amplified if the COI sequence has stop codon(s).

L142: “selected by MEGA.” . Which models? Nucleotide substitution models? Which model(s) was/were chosen? Did MEGA choose the model based on the AIC or BIC criterion? In this case, two different models were chosen by the two criteria, which one did MEGA or you select? For such a shallow phylogeny, it may be OK not to list this. On the other hand, attention to detail matters. (OK I see you put it in the results section. Good!)

L154: “as well as randomly selected sequences”. What do you mean by randomly selected? Why would you ever randomly select sequences for a phylogeny?

L173: “compared because of high variability” . High variability of “what”?

L178: “recorded Chthamalid” . Write chthamalid

L190: “and Chthamalus Alani” . alani

L194: “Chthamalus proteus” . C. proteus

L221: Chthamalinae has been abandoned. See Chan et al. (2021)

L321: “Examination of arthropodal characters”. This is too vague. Define the characters more clearly.

L351: “sea from the eastern coast of”. Eastern

L350-367: This section is not Results but Discussion. Please relocate. I see that you have united results and a short discussion under each species. It may be OK. Consult the editor. I think a single Discussion section is better.

L352: “This hypothesis could be supported by the dispersal of lecithotrophic larvae across” . Chthamalid barnacles have planktotrophic nauplii that develop over 2-3 weeks (Yan and Chan JMBA UK). In fact, had the development been lecithotrophic, we would not expect “vast dispersal”.

L358: “and west Indian”. West Indian. Check this carefully throughout the MS

L359: “adjacent sea”. What do you mean? “seas”?

L385: “The populations were therefore, these populations were grouped under C. cf. challengeri”. Please rephrase

L412: “along the southern and eastern coasts”. Southern and Eastern

L443: “There are clades that cluster species from different groups; for example, C. stellatus and C. montagui on the same clade.” Please rephrase. “Clades cluster species from different groups on the same clade”. Slightly confusing.

L453: “Chthamalu. montagui” . Chthamalus montagui.

L455: “that the “challengeri”. Italicize challengeri

TABLES in general:
I would convert this table into a pair-wise genetic mismatch frequency table. That way the authors can more easily inspect intra- and interspecific thresholds and cut-off values between and within species.

Are the pairwise genetic distances between species significantly different from each other? Maybe this is not the best way to study species delimitation. I am no expert in this field.

TABLE 1:

I would convert this table into a pair-wise genetic mismatch frequency table. That way the authors can more easily inspect intra and interspecific thresholds and cut-off values between and within species.

C. barilani is not in italics in the first row

Is “values” = “differences”?

TABLE 2:

Please define abbreviations

FIGURE 1:

This photo plate could have been more aesthetic. The images have to align and the frames have to be horizontal/vertical and not “skewed” as here. The scale bars are “floating” in different places in the pictures. I suggest placing them at equal distances from the picture frames. The coloring of the background could have been optimized for Fig 1 C and D. There are still some white margins here and there. Maybe give these two images another round of polishing?

FIGURE 2: Please spell out WIO. Maybe mention the total bp alignment that was used to construct this tree.

Did the authors try other mitochondrial or even nuclear markers? I am surprised to see the low support values of COI at this taxonomic scale. Maybe 16S or even 28S and a concatenated alignment of either COI+NaKA or COI+NaKA+16S+28s would be needed to fully resolve the sister relationships of the clades? This is not a “make or break” requirement for publication but would surely be nice.

The authors could insert the pictures of the species on the phylogeny. Would be a bit easier to appreciate their differences then. It seems there is space enough around the clades.

FIGURE 3:

Please insert the legend from Fig 2 as well. Difficult to scroll back to Fig 2 for checking which annotations belong to which species

FIGURE 4:

Suggest making the frames of equal size. This can be done by inserting a white line of equal size between the aligned images. The 0.5 and 3.0 text pieces appear to me as different fonts. Please ensure that the same font is used throughout. The lower right corner of Fig 4B has not been darkened entirely. Suggest painting that area black. There is a tiny piece of black that exceeds the width of the figure in Fig C. Suggest trimming the edges of the figure.

FIGURE 5:

Was the same font as in other figures used? Suggest standardizing font sizes and types.

FIGURE 6:

The scale bar outline is different in this figure from other figures. Suggest standardizing how the scale bar is drawn.

FIGURE 7:

Why are there extra white scale bars in A, B, C, and F? What happened to F? The lower right corner of F does not look so good neither does the gray-scaled scale bar. Suggest standardizing images. The sizes of the frames are not equal. Please optimize this.

FIGURE 8:

The figure letters and scale bars are floating freely around and not at a standardized distance from the edges. Frame and images sizes are not equal. Please optimize this.

FIGURE 9:

It is OK 8 (good in fact) that B was cut in the lower-left corner to harbor C. But please make an effort to standardize frame sizes. The figures look, to me, very sloppy. The scale bars are not of equal size, font, and type. In C and E there are two scale bars.

FIGURE 10:

This is also a sloppy figure and requires substantial edits. Make frame sizes equal. Make sure image edges are aligned. Make sure that scale bars are aligned and of the same width/size. C is thick and white. A and B are thin and black. The scale bar in C is covering the material, but it looks like the scale bar in A was deliberately place above the Cirrus I. Why this putative discrepancy? E covers D without a frame. This should really be optimized with equal frame sizes. Why is there suddenly a black frame around a white scale bar?

FIGURE 11:

Are those numbers indicating support values? I am surprised to see them so low at the “backbone”. Maybe the authors should either attempt to concatenate NaKa and COI or add maybe 16S, 18S, 28S or H3 sequences additionally to get enhanced resolution? Although the species level resolution looks fine, the low support values matter for the bigger picture. I am curious as to why the species assembly shown here was not included in Fig 4? The relationships appear different. Why? This highlights why a separate discussion is warranted.

Experimental design

The experimental design is principally OK. I understand why COI was chosen as the species delimiting marker. I would encourage the authors to argue why NaKa was chosen as a marker over e.g., 16S or a nuclear ribosomal gene? Clearly, COI alone cannot distinguish the backbone of the family, so maybe an extra gene and concatenated trees are needed. Such additional sequencing not “make or break” but would improve the paper. There are balanomorphan specific 16S primers in the literature. John Wares has used EF-1a on Chthamaloids. Maybe it is worth using these primers for also including some of his specimens as outgroup/ingroup?

It is nice to see that both macro, LM, and SEM imaging are being used.

Validity of the findings

The authors resolve the phylogeography of Chthamalid barnacles in the Western Indian Ocean (WIO). However, the potential impacts of the low support values need to be discussed more upfront. I suggest writing a separate discussion section where phylogeographical patterns are discussed in greater depth and also in the light of the dispersal potential of the species (the authors already initiated an interesting discussion on this). The authors already commented on this, which I really liked (although they misinterpreted the development as lecithotrophic, which in fact is planktotrophic).

I would also like to see a deeper discussion on the morphological differences. The authors write several places that “arthropodal characters”, “morphological features” or “morphology” agree with the molecular monophyly of species. What are these characters specifically? It is fine that the species are described in detail but this is done in a telegraphic language that is extremely difficult to compare between species in the text. Since none of the important morphological characters, as listed by the authors, appears on the phylogeny or in a table, it is extremely difficult to compare and assess the differences. I suggest the authors to make a table of key differences in shell, feeding apparatus, trophi and/or other structures that are commonly used to distinguish the barnacles studied. It is at present really difficult to assess how the authors handle differences between gene trees, “morphology” and the somewhat different tree in Fig 11.

The “species delimitation” of WIO species seems to hold and appears valid enough although a broader discussion of the COI+NaKA+morphology approach needs to be further discussed, I think. Many other papers used nuclear and other mitochondrial markers on chthamaloids and come to different genus-level relationships. I am slightly confused by the differences between Fig 4 and Fig 11. I am not an expert in chthamaloids so maybe the authors can explain these differences? I did not fully understand the purpose of Fig 11. The authors write that “However, the conformity between the phylogenetic tree (Fig. 11) and the grouping based on morphological parameters is only partial”. So Fig 11 is intending to challenge the discrepancy between morphology and genetic markers? Or? I do not fully understand how this, particularly when thinking of the overall purpose of the paper which again is to resolve phylogeography of WIO chthamalid barnacles.

I would therefore have liked to see a broader discussion of how different approaches and models may affect the phylogeographic assessment.

Additional comments

Already given above.

Reviewer 2 ·

Basic reporting

In general, this is a well-written manuscript. The figures are appropriate, especially the phylogenetic trees (Figs. 2 and 3) are well-designed. I suggest you add a map with sampling locations and the distribution of all chthamalid species for me and anybody else not familiar with the geographic area.
The english is sufficient. I suggest the authors check spelling before final submission.
However, I find references are missing in several instances, which I marked by “REF” in annotated PDF. I also suggest you include the following relevant literature:
Wares, J. P. (2020). Small, flat, and gray: Cryptic diversity in chthamalid barnacles in the global context of marine coastal biogeography (Cirripedia: Balanomorpha: Chthamalidae). The Journal of Crustacean Biology, 40(1), 1-16.
Wares (2020) points out the gaps in chthamalid phylogeography, which you are filling with this MS

Wares, J. P., Pankey, M. S., Pitombo, F., Daglio, L. G., & Achituv, Y. (2009). A “Shallow Phylogeny” of Shallow Barnacles (Chthamalus). PLOS ONE, 4(5), e5567.

Chan, B. K. K., Corbari, L., Rodriguez Moreno, P. A., & Tsang, L. M. (2017). Molecular phylogeny of the lower acorn barnacle families (Bathylasmatidae, Chionelasmatidae, Pachylasmatidae and Waikalasmatidae) (Cirripedia: Balanomorpha) with evidence for revisions in family classification. Zoological Journal of the Linnean Society, 180(3), 542– 555. https://doi.org/10.1093/zoolinnean/zlw005

Wares et al. (2009) and Chan et al. (2017) reconstruct chthamaid phylogenies, and their results should be included into the phylogenetic discussion.

I have made some additional remarks directly into the pdf of the MS, which you should consider.

Experimental design

The experimental design is good. Sample sizes are adequate. The morphological analysis is excellent. The use of two genetic markers, one mitochondrial and one nuclear, is also recommendable.

Validity of the findings

The findings are overall valid. The phylogenetic discussion is, in my opinion, the weakest point. On the one hand, large scale phylogenetic reconstructions exist (Wares et al. 2009, Chan et al. 201). On the other hand, this discussion point seems out of place for this manuscript. I would focus on biogeographic or phylogeographic inferences that can be drawn from your data.

Additional comments

NA

Annotated reviews are not available for download in order to protect the identity of reviewers who chose to remain anonymous.

---

## Round 0.2 · accepted · Accept

Thank you for your careful revision and attention to the previous two reviewers. I'm glad you liked both the speed and content of the reviews. I agree that they have helped improve your work. I have gone through your response and found your adjustments satisfactory. I am now happy to recommend acceptance. Congratulations.